# Exploring Data Scaling Trends and Effects in Reinforcement Learning from Human Feedback

**Wei Shen, Guanlin Liu,**[*] **Yu Yue, Ruofei Zhu, Qingping Yang, Chao Xin, Lin Yan**
ByteDance Seed
{shenwei.0917, guanlin.liu, yueyu, zhuruofei.2457,
jiangyun.jy, xinchap.xc, neil}@bytedance.com

## Abstract

Reinforcement Learning from Human Feedback (RLHF) is essential for aligning large language models (LLMs) with human preferences and values. While recent research has primarily focused on algorithmic advancements—such as reducing computational overhead or strengthening reward models to mitigate reward hacking—the critical role of prompt-data construction and its scalability has received comparatively less attention. In this paper, we address this gap by systematically exploring data-driven bottlenecks that currently hinder RLHF performance scaling, focusing specifically on the challenges posed by reward hacking and decreasing response diversity. To mitigate reward hacking, we introduce a hybrid reward system combining reasoning task verifiers (RTV) and a generative reward model (GenRM). This approach enables accurate assessment of responses against clearly defined ground-truth solutions. Additionally, in order to ensure response diversity and enhance learning effectiveness, we propose a novel prompt-selection method named **Pre-PPO**, explicitly identifying training prompts that are inherently challenging and thus less prone to reward hacking. Furthermore, we find that **prioritizing mathematical and coding tasks during the early phases of RLHF training** significantly boosts performance, given that these tasks naturally encode fine-grained response distinctions and possess clearly defined ground truths. Through experiments conducted on both small and large models, we demonstrate the effectiveness and scalability of our proposed methods. Our approach exhibits robust generalization capabilities, especially on challenging and out-of-distribution tasks, while yielding significant improvements in mathematics-intensive (STEM) and coding domains. Moreover, the proposed strategies enable the model to effectively capture subtle, task-specific distinctions during the RLHF process, substantially enhancing overall model performance. This work emphasizes the critical role of careful data construction and provides practical methodologies for addressing key performance bottlenecks in RLHF.

## 1 Introduction

Reinforcement Learning from Human Feedback (RLHF) is a crucial technique for aligning large language models (LLMs) with human values and preferences [1, 2, 3, 4, 5]. Recently, numerous studies have focused on enhancing the efficiency and performance of RLHF at the algorithmic level. Methods proposed include dropping the critic model from PPO to reduce computational overhead [6, 7, 8] and filtering noisy samples during the PPO sampling process to achieve more efficient training and improved performance [9]. Additionally, many papers [10, 11, 12] focus on leveraging the capability of the reward model to mitigate the reward hacking problem, thereby enhancing the

---

[*]Corresponding Author. Email: guanlin.liu@bytedance.com

39th Conference on Neural Information Processing Systems (NeurIPS 2025).

performance of RLHF. However, there have been few studies [13] focusing on the construction of RLHF data (i.e., training prompts) and its performance scaling based on these training prompts, despite extensive research on large-scale datasets during pretraining and supervised fine-tuning (SFT) stages [14, 15, 16].

In this paper, we find that simply scaling the number of training prompts during RLHF does not lead to performance improvements. Accordingly, we investigate the bottlenecks of data scaling in RLHF and propose novel methods for constructing training prompts, as well as strategies to enhance RLHF performance. We identify two primary obstacles to effective data scaling: reward hacking and the deterioration of model response diversity. To mitigate reward hacking, we develop a hybrid reward system that integrates Reasoning Task Verifiers (RTV) [17, 18] with Generative Reward Models (GenRM) [19, 20], allowing model predictions to be validated against reliable ground-truth references. Additionally, we find that training models primarily on coarse-grained response differences rapidly diminishes response diversity, causing important fine-grained distinctions among responses to be overlooked. While directly preventing this reduction in diversity during RLHF training remains challenging, we demonstrate that targeted prompt selection can facilitate the learning of subtle, fine-grained distinctions and make reward hacking more difficult. Specifically, we introduce an innovative Pre-PPO prompt selection method that emphasizes more challenging prompts and is inherently more robust to reward hacking. Furthermore, we find that initially prioritizing mathematical and coding tasks during RLHF training consistently yields superior results, as these tasks naturally encourage fine-grained distinctions and provide safeguards against reward hacking due to their well-defined ground truths.

**Our main contributions are as follows:**

- **Comprehensive Analysis of RLHF Scaling Bottlenecks:** We systematically analyze the key factors that hinder RLHF performance scaling, identifying *reward hacking* and the *deterioration of model response diversity* as primary obstacles.

- **Joint Prompt Selection and Task Prioritization Strategy:** We propose a novel training strategy that combines a **Pre-PPO** prompt selection method—which strategically chooses difficult and informative prompts—with a curriculum that **prioritizes mathematical and coding tasks in the early stages of RLHF training**. This unified approach enables the model to acquire fine-grained distinctions while improving robustness to reward hacking.

- **Empirical Validation Across Model Scales:** We conduct extensive experiments on two distinct model sizes, demonstrating the effectiveness and scalability of our proposed methods. Comprehensive ablation studies confirm that each technique individually contributes to RLHF performance improvements at different model scales.

## 2 Related Work

**Reward hacking.** Reward hacking occurs when an AI system exploits flaws in the reward function to maximize rewards without achieving the intended objectives. Consequently, the success of RLHF heavily depends on the quality of the reward model. Unfortunately, reward models often struggle to provide accurate scores due to three main challenges: 1) mis-specified reward modeling in representing human preferences [21, 22]; 2) the presence of incorrect and ambiguous preferences in training datasets [23, 24]; and 3) poor generalization ability [25]. These inaccuracies in reward modeling have been identified as major contributors to reward hacking and hallucination in LLMs [26]. Recent work by Zhang et al. [20] introduced a generative reward model (GenRM) to validate model predictions against ground-truth responses, demonstrating greater resistance to reward hacking and has been adopted by state-of-the-art LLMs such as DeepSeekV3 [27]. Additionally, Deepseek-R1 [18] developed reasoning task verifiers (RTV) that have proven effective in addressing reward hacking, particularly in mathematical, coding, and other reasoning tasks. While previous research has focused on improving the accuracy of reward models themselves, our work takes a different approach: we aim to design an effective RLHF data construction method under a robust reward system that combines both GenRM and RTV to mitigate the reward hacking problem.

**RLHF data construction.** There are few works that focus on how to construct RL data (i.e., RL training prompts) to enhance RLHF performance. Gao et al. [28] propose a principled data selection method for the DPO algorithm, where they find overly difficult data hinder alignment and filter out such challenging instances during DPO training. Additionally, Li et al. [29] introduce a strategic

selection method to identify key training prompts from a complete prompt set, achieving comparable RLHF performance while using only a subset of the data. While these methods demonstrate that careful dataset curation can match or exceed the performance of training on complete datasets, there remains a significant gap in understanding the factors that limit RL data scaling in PPO-based training. Specifically, no existing work has systematically analyzed how to select and structure training prompts to substantially improve model performance when using the PPO algorithm.

**RLHF performance scale analysis.** Recent studies have increasingly focused on analyzing RLHF [13, 24, 30], particularly examining its generalization ability and response diversity. Kirk et al. [31] demonstrate that RLHF exhibits superior generalization compared to Supervised Fine-Tuning (SFT) on novel inputs, especially as the distribution shift between training and testing data increases. However, they also observe that RLHF significantly reduces output diversity compared to SFT across various metrics, suggesting a fundamental trade-off between generalization and diversity in current LLM fine-tuning approaches. Furthermore, several recent works [32, 33] investigate how RLHF can effectively distill the best responses as evaluated by reward models, proposing various algorithms to enhance this distillation capability. In our study, we similarly observe that the diminishment of response diversity impedes RLHF scaling, particularly when models attempt to learn coarse differences among responses. Additionally, we observe that RLHF performance is only comparable to the strategy of sampling five responses from the SFT model and then selecting the highest-ranked one according to the reward model (i.e., SFT Bo5). This observation motivates further investigation into methods for enhancing the effectiveness of RLHF.

## 3 Approach

### 3.1 Framework Overview

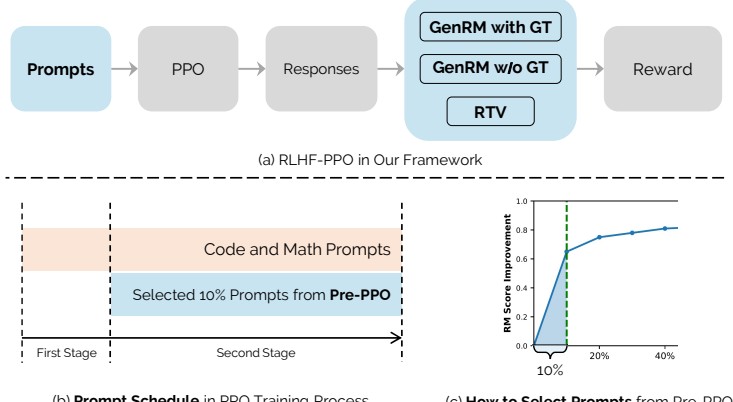

(a) RLHF-PPO in Our Framework

(b) **Prompt Schedule** in PPO Training Process

(c) **How to Select Prompts** from Pre-PPO

Figure 1: **Overview of our RLHF framework.** Our pipeline consists of two stages: (1) *Reward Model Training*, where we construct three complementary reward models: a Bradley-Terry (BT) model trained on human pairwise preferences; a Generative Reward Model (GenRM) assigning explicit reward scores using either ground-truth (GT) responses or BT-best responses when GT is unavailable; and Reasoning Task Verifiers (RTV) providing task-specific validation (such as code-execution sandboxes); and (2) *Reinforcement Learning Optimization*, where the language model is trained using PPO guided by GenRM and RTV feedback, and leveraging strategically selected prompts via our Pre-PPO method. Additionally, we explicitly prioritize mathematical and coding prompts during early training stages, subsequently incorporating general-domain tasks.

As shown in Figure 1, our RLHF pipeline consists of three main stages:

- **Initial Supervised Fine-tuning**: We first fine-tune the pre-trained language model on human-written demonstrations to achieve basic instruction-following capabilities.

- **Reward Model Training**: We prepare three types of reward models: the Bradley-Terry Reward Model (BT Model), the Generative Reward Model (GenRM), and Reasoning Task Verifiers (RTV). The BT model learns reward functions using pairwise comparison data, optimizing parameters

via maximum likelihood estimation to infer underlying reward scores from human preferences. The Generative Reward Model (GenRM) is trained using Pairwise Reward Modeling (pairRM), where the model learns from human preference judgments of paired outputs [34]. Rather than assigning individual scores to each output, GenRM directly predicts a comparative score for each pair, optimizing these pairwise comparisons to align closely with human evaluations. To provide GenRM with a reliable ground truth for comparison, we collect explicit ground-truth answers for reasoning tasks. For other task types, we utilize the trained BT reward model to select the best outcome from N candidate samples generated by the supervised fine-tuned (SFT) model. For RTV, we construct a series of specialized verifiers to directly validate the correctness of model responses for specific tasks. These include, for example, code sandboxes for programming tasks, which can execute and evaluate code outputs in real-time.

- **Reinforcement Learning Optimization**: We leverage a combination of GenRM and RTV to provide comprehensive feedback for optimizing the language model through PPO. This process is guided by carefully curated training prompts and a well-crafted training strategy. The iterative nature of this approach progressively refines the model's outputs by maximizing predicted reward scores while ensuring minimal deviation from its original policy.

During RL training, we observe two key factors that hinder the performance scaling of RLHF: first, **reward hacking**. As shown in Figure 2, the overall performance of the model undergoing RLHF exhibits an initial increase followed by a decline during training. Specifically, abilities in mathematics, creative tasks, and instruction following all follow this pattern of improvement and subsequent deterioration. Our analysis reveals that reward hacking occurs across these tasks, where the model learns to generate responses containing certain syntactic patterns that artificially inflate reward scores rather than genuinely improving task performance; second, **the deterioration of response diversity**. During the RLHF process, we observe a continuous decline in the entropy of model responses (illustrated in subfigure (a) of Figure 6 in the Appendix), indicating a decrease in response diversity.

Despite numerous efforts to address reward hacking and diminishing response diversity—including approaches such as iterated RLHF [24, 35] and reinforcement learning from pre-trained models [18, 36]—these challenges remain difficult to fully resolve. To mitigate these issues, we propose a proactive prompt selection strategy combined with a curriculum learning approach.

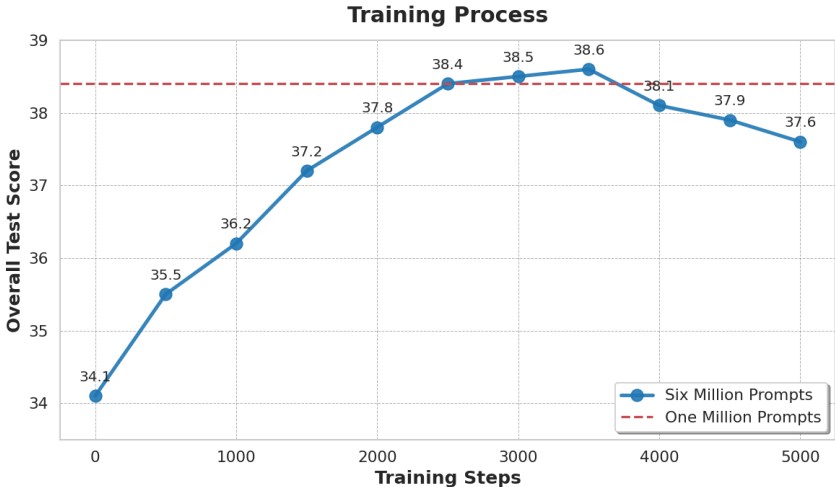

Figure 2: Overall test scores from the initial run using an expanded dataset combining newly collected data (six million prompts) with the original dataset (one million prompts). Despite increasing dataset size substantially, RLHF did not yield improvements in performance. Additionally, the best performance was observed at around the 3,500-step mark, after which test scores gradually declined.

## 3.2 Pre-PPO for Training Prompts Selection

**Initial PPO Experiment.** In our first trial, we initially collected 5 million new training prompts covering various domains, such as mathematics, coding, creative writing, and additional tasks. These new prompts were combined with the original 1 million prompts to train for the first trial. As illustrated in Figure 2, we observed that the RLHF performance did not improve despite the increase in the number of prompts. Consequently, we conclude that simply expanding the number of training prompts does not necessarily yield improved RL performance.

**Reward Analysis of Newly Collected Prompts.** We investigated why newly collected prompts did not improve RLHF performance by analyzing their reward scores. We found that approximately 90% of these prompts received reward scores greater than 0.5 on a scale from 0 to 1 (see Appendix Figure 7). In this scoring scheme, a score of 0.5 indicates that the model's output is comparable to the reference, while scores above 0.5 suggest superior performance. Our GenRM is trained to compare the model response with the ground truth in reasoning tasks and SFT Best-of-N responses in other tasks. Therefore, scores above 0.5 imply that the model-generated outputs were judged as superior to these presumed optimal responses. However, after careful manual inspection, we discovered that a substantial portion of these high-scoring outputs exhibited reward hacking behavior and were qualitatively worse than the original best-selected responses.

**Selecting Prompts with Lower Reward Model Scores for RL Training.** Given the observations above, we designed a selection algorithm called **Pre-PPO**, which explicitly identifies prompts with lower reward model scores for use in the initial PPO experiment. These low-scoring prompts are both more challenging for the model to learn from and less susceptible to reward hacking. Finally, we combined these selected prompts with the original prompt dataset to retrain the RL model. Recognizing that reward model scores exhibit distinct distributions across different task domains, we normalize these scores within each domain before performing prompt selection.

## 3.3 Early-stage RLHF: Prioritizing Mathematical and Coding Tasks

In our initial trials, we observed that test scores for both coding and math tasks steadily improved throughout training. In contrast, other tasks without explicit ground-truth encountered reward hacking issues early in the RLHF process (see Appendix Figure 10). We attribute the improvement in coding and math tasks to their evaluation methods: specifically, these tasks are assessed by RTV and GenRM using ground-truth references, which makes them inherently more resistant to reward hacking. Accordingly, we explicitly trained the RLHF model on math and coding prompts during the early stages. Subsequently, we combined these math and coding tasks with general-domain prompts to continue RLHF training. This approach can enhance performance on both coding and math tasks while preserving general capabilities.

# 4 Experiments Details

## 4.1 Experimental Setup

We describe our experimental setup as follows:

- **Models:** We conducted our experiments using two pre-trained Mixture-of-Experts (MoE) language models: a smaller model (approximately 25B total parameters) and a larger model (approximately 150B total parameters). Due to the MoE architecture, only about 10% of the total parameters are activated per token.

- **Prompts:** Our original dataset consists of one million training prompts collected from publicly available sources and through human annotation. These prompts span diverse domains, including mathematics, coding, instruction-following, creative writing, logical reasoning, and other related tasks. To expand our dataset, we additionally collected five million new prompts primarily from open-source resources, primarily covering mathematics, coding, instruction-following, and creative writing tasks. (The detailed distribution of our training prompts is illustrated in Figure 9 in the Appendix.)

- **Evaluations:** Given that external public benchmarks are susceptible to data contamination, we constructed an internal evaluation set that comprehensively covers multiple skill areas. This set

| Method | LR | IF | STEM | Coding | NLP | Knowledge | CU | OOD | Overall |
|---|---|---|---|---|---|---|---|---|---|
| Initial-Run-25B (V1.0) | 27.1 | 34.8 | 49.3 | 51.6 | 24.7 | 37.0 | 40.0 | 39.0 | 37.7 |
| Baseline-25B (V1.0) | 26.4 | 35.1 | 48.8 | 50.9 | 24.8 | 36.1 | 40.6 | 40.5 | 37.7 |
| Data Scale-25B (V1.0) | 28.7 | 36.1 | 50.4 | 53.3 | 24.2 | 36.6 | 39.7 | 43.6 | 38.8 |
| Improvement | **+2.4** | +1.1 | **+1.6** | **+2.4** | -0.6 | +0.6 | -0.9 | **+3.1** | **+1.1** |
| Baseline-150B (V1.0) | 37.3 | 46.3 | 55.6 | 55.5 | 45.7 | 46.8 | 58.4 | 54.5 | 49.7 |
| Data Scale-150B (V1.0) | 39.6 | 46.0 | 56.5 | 58.7 | 44.9 | 47.9 | 59.6 | 55.6 | 50.8 |
| Improvement | **+2.2** | -0.4 | +0.9 | **+3.2** | -0.8 | +1.1 | +1.2 | **+1.2** | **+1.1** |
| Baseline-25B (V2.0) | 17.6 | 26.5 | 26.5 | 41.2 | 21.2 | 28.2 | 19.6 | 21.3 | 23.9 |
| Data Scale-25B (V2.0) | 19.9 | 27.3 | 29.5 | 42.3 | 21.8 | 28.9 | 20.2 | 21.7 | 25.1 |
| Improvement | **+2.3** | +0.8 | **+3.0** | **+1.1** | +0.6 | +0.7 | **+0.8** | +0.4 | **+1.2** |
| Baseline-150B (V2.0) | 29.5 | 36.3 | 28.0 | 48.5 | 29.5 | 45.6 | 36.8 | 35.0 | 34.0 |
| Data Scale-150B (V2.0) | 31.2 | 36.4 | 31.9 | 50.7 | 32.3 | 45.5 | 36.6 | 37.1 | 35.4 |
| Improvement | **+1.8** | +0.1 | **+3.9** | **+2.1** | **+2.7** | -0.1 | -0.2 | **+2.1** | **+1.4** |

Table 1: We present a performance comparison between our proposed method, termed 'Data Scale,' and a baseline method (PPO-based RLHF) on evaluation datasets V1.0 and V2.0. Results are reported across various abilities, including logical reasoning (LR), instruction-following (IF), STEM tasks, coding, natural language processing (NLP), knowledge, contextual understanding (CU), and out-of-distribution generalization (OOD). Results highlighted in **bold** indicate statistically significant improvements.

consists of two main versions, V1.0 and V2.0, as well as a separate human evaluation subset. While V1.0 and V2.0 share some overlapping prompts, the newly introduced prompts in V2.0 are notably more challenging. Importantly, all prompts in V2.0 and the human evaluation set are not accessible to researchers at any stage during optimization, ensuring an unbiased and fair assessment. Model performance was assessed using both automated (LLM-based) and manual (human-based) evaluations, with the latter conducted using the newly introduced human evaluation subset. Furthermore, to ensure fair comparison with external models and reproducibility of our method, we conduct comprehensive evaluations on a series of open-source benchmarks in Appendix B.

Further details on experimental setups—including *Experimental Details of Pre-PPO* and *Experimental Details of Prioritizing Mathematical and Coding Tasks*—are provided in the Appendix C.2.

## 4.2 Experimental Results

**Overall Evaluation Results.** The experimental results presented in Table 1 and Table 2 demonstrate the following key findings:

- **Overall Performance Improvement.** Our proposed approach (combining Pre-PPO with prioritized mathematical and coding tasks) consistently and significantly outperforms the baseline method (PPO with the original dataset) across different model sizes and evaluation datasets.

- **Strong Generalization on More Challenging Test Sets.** We evaluate checkpoints from both the baseline method and our proposed approach at every 100 training steps using TestSet V1.0, and select the best-performing checkpoint from training steps up to 4000. Under this evaluation, our approach achieves a noticeable improvement (**+1.1**) over the baseline on TestSet V1.0. Furthermore, when comparing the best checkpoints from each method on a more challenging TestSet V2.0, our approach yields an even greater performance increase (**+1.4**). Given that TestSet V2.0 contains substantially more challenging prompts than TestSet V1.0, these results indicate that the proposed approach exhibits robust generalization capability, especially on harder, out-of-distribution tasks.

- **Significant Improvements in Mathematical and Coding Tasks.** Our proposed approach notably enhances performance on mathematics-intensive (STEM) and coding tasks. Specifically, we observe improvements of **+3.9** points in STEM (Large, V2.0) and **+3.2** points in coding (Large, V1.0), alongside consistent gains across other model sizes and datasets. We attribute these significant improvements to our strategic prioritization of mathematical reasoning and coding tasks during the early stages of RLHF training, which effectively strengthens the model's capabilities in these specialized areas.

| Method | Knowledge | STEM | IF | Creation | Coding | Overall |
|---|---|---|---|---|---|---|
| Baseline-Large | 63.3 | 76.7 | 46.7 | 52.1 | 67.2 | 64.4 |
| Data Scale-Large | 66.1 | 80.6 | 48.3 | 54.6 | 71.0 | 67.6 |
| Improvement | +2.8 | **+3.9** | +1.6 | **+2.5** | +3.8 | **+3.2** |
| p-value | 0.41 | 0.04 | 0.39 | 0.09 | 0.12 | 0.01 |

Table 2: Performance comparison based on comprehensive human evaluations between our proposed method, termed 'Data Scale' and the baseline method (PPO-based RLHF). Results are shown across multiple abilities, including Knowledge, STEM, Instruction-Following (IF), Creation, Coding, and Overall performance. Improvements highlighted in **bold** indicate statistically significant differences. All metrics represent aggregated scores from human assessments.

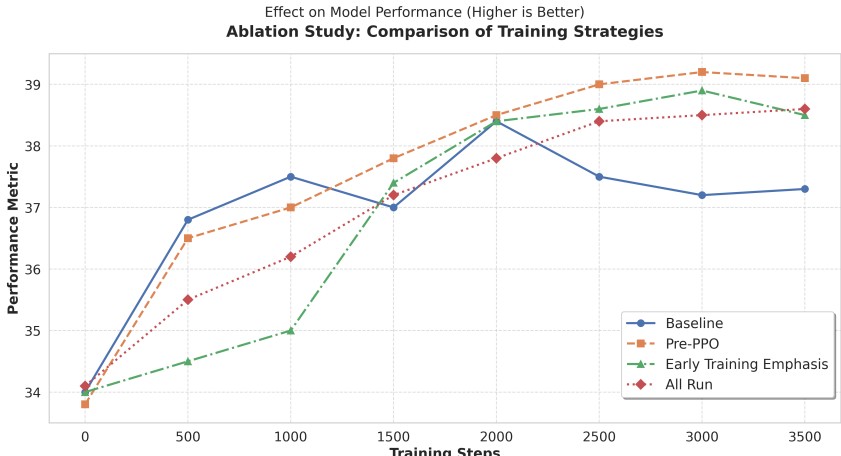

Figure 3: Ablation study on small-size model.

**Case Study Comparisons.** Based on feedback from human annotators (as shown in Table 2), we observe clear improvements across different task categories. For STEM-related tasks, annotators noted notable enhancements in logical reasoning and content richness, as exemplified in Case F.1 of the Appendix. Meanwhile, complex creative tasks exhibited moderate improvements, specifically in adhering to secondary instructions and overall content quality, including enhanced literary style and increased richness, as illustrated in Case F.2. For coding tasks, annotators recognized distinct improvements in information accuracy and depth of content (Case F.3).

### 4.3 Ablation Studies

To investigate the impact of Pre-PPO and of the early-stage emphasis on mathematical and coding tasks, we independently compare the performance of each approach against baseline methods on TestSet V1.0. Due to computational constraints, all subsequent experiments, except for those analyzing scaling trends with respect to model size, are conducted exclusively using the small-sized model.

**Prompt Selection with Pre-PPO.** The experimental results depicted in Figure 3 reveal that the Pre-PPO method achieves comparable performance to the baseline approach up to the 2000-step mark in the training process. Notably, Pre-PPO demonstrates continued improvement between 2000 and 4000 steps, whereas the baseline performance plateaus. This sustained enhancement suggests that the prompts selected through Pre-PPO are more resistant to "hacking", thereby fostering continued learning and ultimately boosting the effectiveness of RLHF.

**Impact of Early Training Emphasis on Mathematical and Coding Tasks.** As shown in Figure 4, early training emphasis on mathematical and coding tasks can improve the overall performance of RLHF. Furthermore, as shown in Figure 4, early emphasis on mathematical and coding tasks during training significantly enhances both the coding and STEM performance of RLHF models. Notably, in the coding task, the early training emphasis method surpasses the baseline performance plateau

| Method | Logical Reasoning | IF | STEM | Coding | NLP | Knowledge | CU | OOD | Overall |
|---|---|---|---|---|---|---|---|---|---|
| Baseline-Large (V2.0) | 29.5 | 36.3 | 28.0 | 48.5 | 29.5 | 45.6 | 36.8 | 35.0 | 34.0 |
| Pre-PPO-Large (V2.0) | 31.3 | 35.9 | 30.8 | 49.5 | 32.3 | 45.7 | 36.1 | 37.9 | 35.1 |
| Improvement | **+1.8** | -0.4 | **+2.5** | **+1.0** | **+1.8** | +1.1 | -0.7 | **+2.9** | **+1.1** |
| Data Scale-Large (V2.0) | 31.2 | 36.4 | 31.9 | 50.7 | 32.3 | 45.5 | 36.6 | 37.1 | 35.4 |
| Improvement on Pre-PPO | -0.2 | +0.5 | **+1.1** | **+1.2** | +0.0 | -0.2 | +0.5 | **-0.8** | +0.3 |

Table 3: Ablation Study: Performance Scaling of Pre-PPO and Early Training Emphasis in Large Language Models

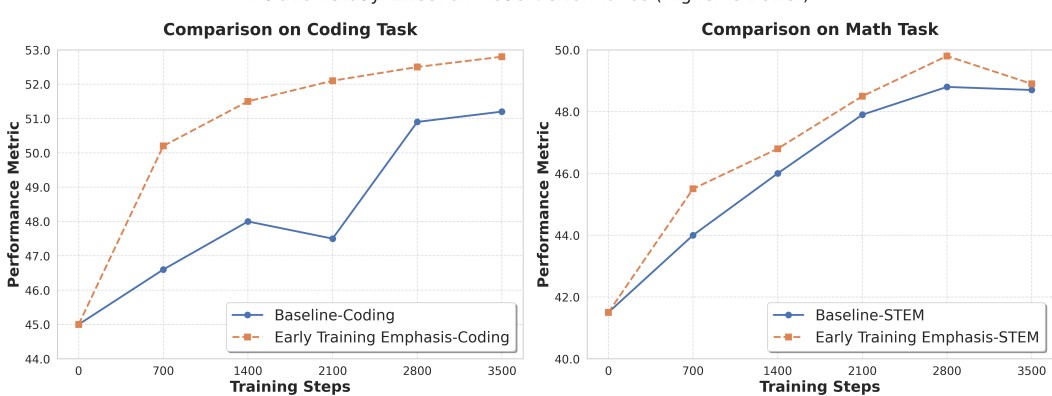

Figure 4: Early emphasis on mathematical and coding tasks significantly improves RLHF performance in both coding and STEM areas on Testset-V1.0. Notably, the coding performance with this approach surpasses the baseline within just 1000 training steps.

as early as the 1000-step mark. This improvement demonstrates the efficacy of prioritizing these foundational skills in the initial stages of training, leading to accelerated learning and superior overall performance.

**Performance Scaling Trends of Model Size.** As shown in Table 1, our method demonstrates direct applicability to large-scale models, achieving significantly better performance than the baseline. This indicates a positive scaling trend with respect to model size.

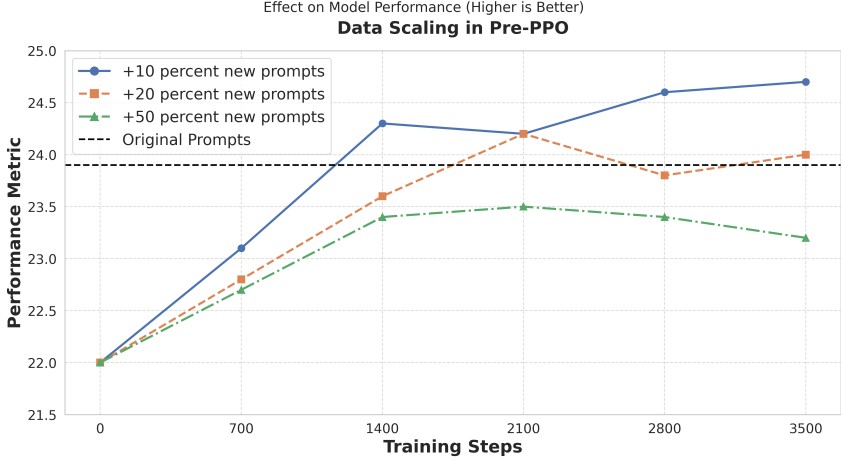

Figure 5: Impact of data scaling on Pre-PPO strategy performance. The graph shows the overall RLHF performance as the percentage of newly collected training data increases from 10% to 20% and 50%.

Furthermore, we investigated the individual performance of Pre-PPO and early emphasis strategies on mathematical and coding tasks when applied to large models. Due to computational constraints, we focused on applying the Pre-PPO strategy (using data selected from the small-size model) to the large model for dataset selection in our ablation study. As shown in Table 3, Pre-PPO yields a significant performance improvement in large models. Similarly, emphasizing mathematical and coding tasks early in the training process of large models results in substantial performance gains in both STEM and coding evaluations. However, this approach yields only marginal improvements in overall performance. Notably, directly scaling the hyperparameters from the small model to the large model does not necessarily yield optimal results; in future work, we will further investigate how to properly scale hyperparameters to ensure even greater performance boosts.

**Data Scaling in the Pre-PPO Strategy.** We investigated the effect of data scaling within the Pre-PPO strategy, as shown in Figure 5. However, increasing the amount of newly collected training data from 10% to 20% or 50% leads to a noticeable degradation in overall RLHF performance. This counter-intuitive result suggests that high-quality training prompts are scarce in real-world settings, and simply scaling the quantity of collected data does not necessarily lead to improvements. In future work, we will explore approaches to generating prompts directly from large language models (LLMs) themselves, which we consider a more promising direction than relying purely on real-world collections.

# 5 Discussion

Although we have demonstrated the effectiveness and positive scaling trend of our method, we aim to further explore the underlying mechanisms that contribute to its success. In Appendix D, we design a series of experiments to seek to understand why our approach enhances the performance of RLHF and how it breaks through two critical bottlenecks: reward hacking and the deterioration of model response diversity. We observed that **early acquisition of fine-grained response differences enhances performance scaling ideally**. Specifically, We find that explicitly prioritizing prompts exhibiting fine-grained response differences—especially in mathematical and coding tasks supervised by reward models with ground-truth or verification feedback—significantly enhances model performance and scalability. In contrast, prematurely introducing tasks dominated by coarse-grained variations (e.g., creative writing) negatively affects learning effectiveness. This highlighted sensitivity underscores the importance of initially acquiring fine-grained response distinctions to mitigate adverse impacts from early exposure to coarse-grained patterns.

Our method requires the collection of a large number of prompts paired with high-quality reference answers, which can be resource-intensive. Additionally, incorporating pre-PPO steps involves executing an initial PPO process to filter and augment data, thereby increasing computational overhead during training. While emphasizing fine-grained response distinctions has shown promise, further research is necessary to understand how best to implement this in different contexts and tasks. Moreover, the impact of our approach on long-form Chain-of-Thought RLHF remains uncertain, as current analyses are limited to specific scenarios and may not directly transfer to more complex, multi-step reasoning tasks. These factors could pose challenges for practical deployment and broader applicability. Please refer to Appendix E for additional discussion.

# 6 Conclusion

In this paper, we explored two key bottlenecks preventing effective data scaling in Reinforcement Learning from Human Feedback (RLHF): reward hacking and declining diversity of model responses. To address these issues, we introduced a hybrid reward system combining Reasoning Task Verifiers (RTV) and Generative Reward Models (GenRM) with ground-truth supervision, significantly enhancing resistance to reward hacking. Furthermore, we proposed a novel Pre-PPO prompt selection strategy that prioritizes challenging prompts to help models better capture fine-grained response distinctions, and demonstrated the advantages of prioritizing mathematical and coding tasks early in the training process. Our results indicate that strategically curating training prompts markedly improves response diversity and overall RLHF performance and scalability. We hope our work provides foundational insights for future research into optimized RLHF data construction and stronger alignment methodologies.

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

# A    The deterioration of model response diversity

During the RLHF training process, we observe a continuous decline in the entropy of model responses (illustrated in subfigure (a) of Figure 6), indicating reduced response diversity. Such a decline not only constrains the model's capability to produce varied and creative outputs but may also negatively impact its adaptability and generalization across diverse tasks and contexts. Additionally, we analyze the entropy across various task categories and observe that tasks associated with creative writing, role-play, and others supervised by GenRM without ground truth exhibit notably higher entropy than tasks involving mathematical, coding, and logical reasoning skills—tasks that typically are supervised by GenRM with ground truth. We compare response entropy between the baseline and our proposed method in subfigures (b), (c), and (d), categorizing the results according to the reward model types: GenRM with ground truth, GenRM without ground truth, and RTV. We observe that, for tasks supervised by GenRM with ground truth or RTV, the response diversity using our method is lower than that of the baseline. In contrast, for tasks supervised by GenRM without ground truth, our method exhibits higher response entropy compared to the baseline. These observations indicate that our proposed method effectively guides the model to focus more explicitly on tasks supervised by RTV and GenRM with ground truth, thus enabling the model to acquire more fine-grained response distinctions during RLHF training.

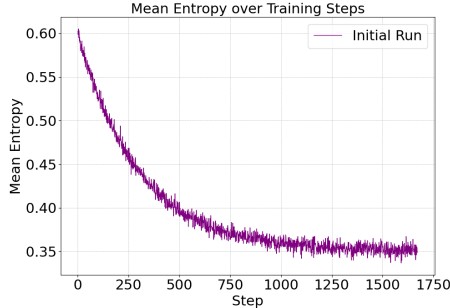

(a) Response entropy change during the RLHF training process.

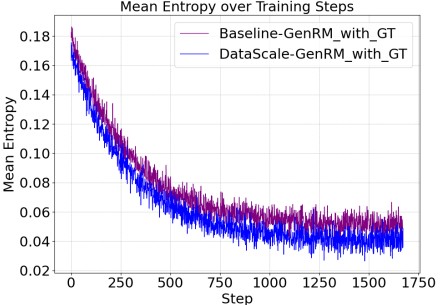

(b) A comparison of response entropy changes during the RLHF training process, aggregated across tasks supervised by the 'GenRM with ground truth'.

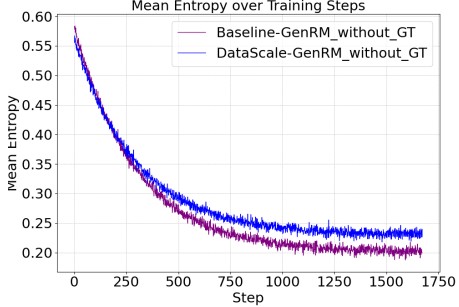

(c) A comparison of response entropy changes during the RLHF training process, aggregated across tasks supervised by the 'GenRM without ground truth'.

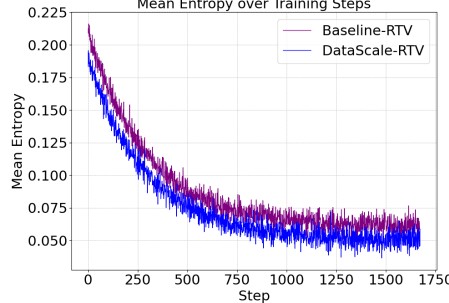

(d) A comparison of response entropy changes during the RLHF training process, aggregated across tasks supervised by the RTV'

Figure 6: The comparison of response entropy change during the RLHF training process

# B    Experimental Results on Open-Source Benchmarks

In this section, we provide a performance comparison between the baseline model and our method. As shown in Table 4, we observe the following patterns:

Table 4: Performance comparison across different task series

| Task Series | Sub-Task Name | Question Count | Baseline Score | Data Scale-Large |
|---|---|---|---|---|
| MATH - Aligned | AMC23 | 40 | 68.8 | 68.7 |
| | MATH500 | 500 | 88.2 | 87.8 |
| AIME (Open-Style) | AIME 2025 | 30 | 18.6 | 17.9 |
| | AIME 2024 | 30 | 26.9 | 27.2 |
| Coding - Aligned | LiveCodeBench | 131 | 30.3 | 36.2 |
| General Knowledge | MMLU | 150 | 87.3 | 87.3 |
| | MMLU pro | 1000 | 79.6 | 80.1 |
| Reasoning | GPQA diamond | 198 | 59.6 | 60.1 |
| | SuperGPQA | 250 | 52.8 | 54.0 |

- **Substantial gains in coding tasks**: LiveCodeBench shows a notable improvement from 30.3 to 36.2, representing a 19.5% relative increase.

- **Consistent performance in general knowledge**: Both MMLU (87.3 → 87.3) and MMLU Pro (79.6 → 80.1) maintain or slightly improve their scores, demonstrating the preservation of broad knowledge capabilities.

- **Marginal trade-offs in standard mathematical tasks**: Minor decreases are observed in AMC23 (68.8 → 68.7) and MATH500 (88.2 → 87.8), though the differences remain within negligible ranges.

- **Comparable performance on challenging mathematical problems**: AIME tasks show mixed but comparable results (AIME 2025: 18.6 → 17.9; AIME 2024: 26.9 → 27.2), indicating robust handling of complex mathematical reasoning.

Overall, our method demonstrates strong improvements in coding capabilities while maintaining competitive performance across mathematical and general knowledge tasks. The results suggest that our approach successfully enhances specialized skills without compromising the model's general capabilities, achieving a well-balanced performance profile across diverse task domains.

## C   Experimental Details

### C.1   Reward Analysis of Newly Collected Prompts.

We investigated why newly collected prompts did not improve RLHF performance by analyzing their reward scores. As shown in Figure 7, We found that approximately 90% of these prompts received reward scores greater than 0.5 on a scale from 0 to 1. In this scoring scheme, a score of 0.5 indicates that the model's output is comparable to the reference, while scores above 0.5 suggest superior performance. Our GenRM is trained to compare the model response with the ground truth in reasoning tasks and SFT Best-of-N responses in other tasks. Therefore, scores above 0.5 imply that the model-generated outputs were judged as superior to these presumed optimal responses. However, after careful manual inspection, we discovered that a substantial portion of these high-scoring outputs exhibited reward hacking behavior and were qualitatively worse than the original best-selected responses. Moreover, we observed a direct correlation between the magnitude of the reward score and the severity and frequency of reward hacking instances. The higher the reward score, the more severe and frequent the reward hacking issue became. This finding reveals a critical limitation in our current reward model and underscores the need for more robust evaluation metrics that can effectively distinguish between genuine improvements and instances of reward hacking.

### C.2   Detail Experiment Setups

**Experimental Details of Pre-PPO:** As shown in Figure 2, we first combine the newly collected prompts with the original prompts to construct the training prompt set for the initial run. Then, as

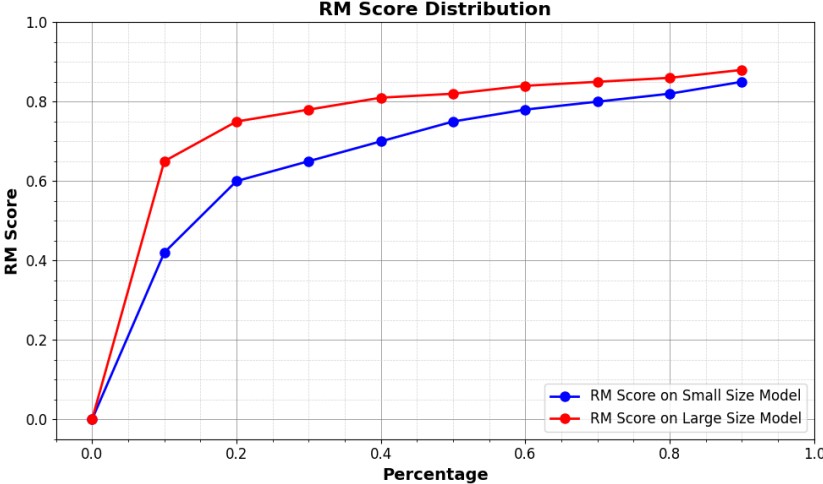

Figure 7: Distribution of reward scores for newly collected prompts. The x-axis shows the percentage of prompts. The y-axis represents the reward score range from 0 to 1, with 0.5 indicating parity with the reference. Approximately 90% of prompts received scores above 0.5 for both small-size and large-size models, suggesting apparent superiority over reference outputs. However, manual inspection revealed that many high-scoring outputs exhibited reward hacking behavior and were qualitatively inferior to the original best-selected outcomes.

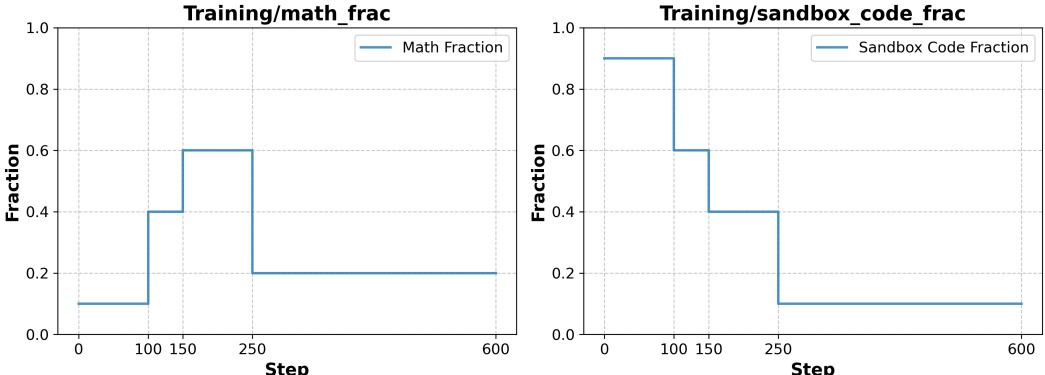

Figure 8: The distribution of prompts across both math and coding task during the training phases

illustrated in Figure 7, we select only the bottom 10% of prompts based on their scores assigned by the generative reward model. This prompt selection process is conducted using the small-sized model. To reduce computational costs, we do not repeat this process on the large-sized model.

**Experimental Details of Prioritizing Mathematical and Coding Tasks.** Since performance on coding tasks is measured via unit tests, which are more robust and less susceptible to reward hacking compared to math tasks, we leverage this property by assigning a higher proportion of coding-related prompts during the early stages of RLHF training. Specifically, we begin training exclusively on coding prompts, gradually introduce mathematical prompts, and ultimately utilize the complete mixed-domain training dataset. The distribution of math and coding prompts throughout the training process is presented in Figure 8.

### C.3 Prompt Distribution

We collect approximately 6 million diverse prompts from open-source resources to construct our RL training prompt set. As illustrated in Figure 9, we categorize these prompts into multiple task types

(e.g., math, knowledge, and creative writing). The relative proportions of each task category within the collected prompt dataset are presented in the figure.

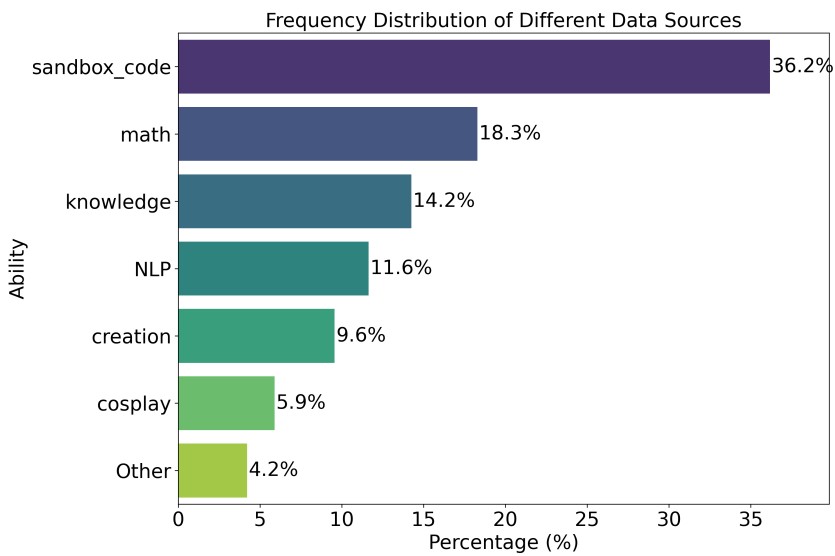

Figure 9: Prompts Distribution covering varies domains.

# D Further Experimental Analysis

Although we have demonstrated the effectiveness and positive scaling trend of our method, we aim to further explore the underlying mechanisms that contribute to its success. Specifically, we seek to understand why our approach enhances the performance of RLHF and how it breaks through two critical bottlenecks: reward hacking and the deterioration of model response diversity.

**Reward Hacking Problems Across Different Reward Models.** Aside from perfect verifiers, any reward model used during RLHF can potentially be hacked. However, as shown in Figure 10, we observe that:

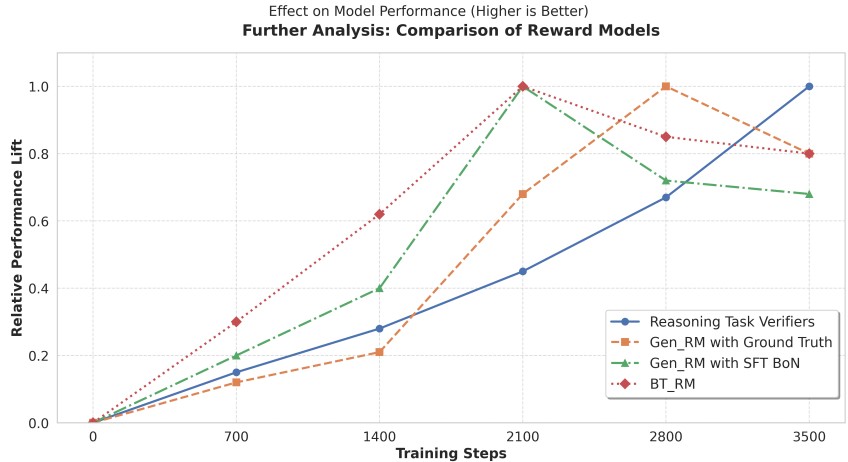

Figure 10: Comparison of Reward Hacking Susceptibility and Performance Trends for RTV, GenRM, and BT Reward Models During RLHF Training

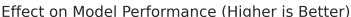

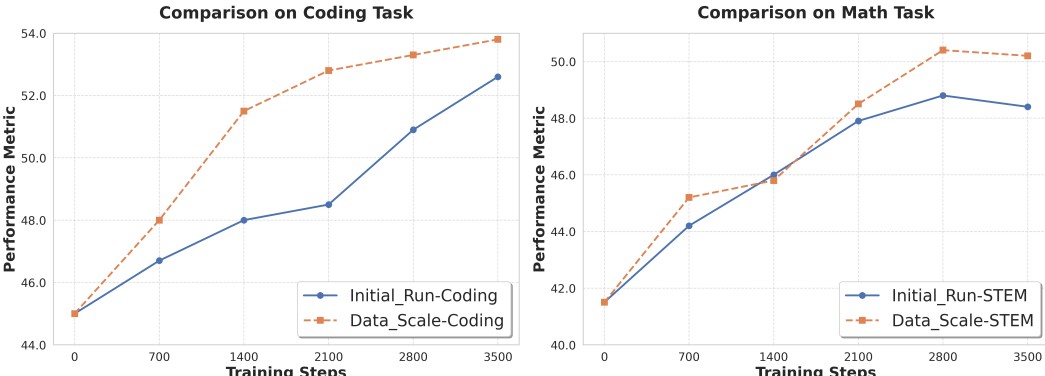

Figure 11: Data Scale method boost both math and code performance.

- For tasks evaluated using RTV, test scores continued to improve throughout the entire RLHF training period. This sustained improvement suggests that RTV provides robust and hack-resistant feedback.

- When using GenRM with ground truth data, we observed consistent score improvements up to approximately the $2800^{\text{th}}$ training step. This indicates that GenRM maintains its effectiveness as a feedback mechanism for a significant portion of the training process.

- In contrast, the BT reward model (or GenRM utilizing responses selected by Best-of-N sampling (BoN) from the SFT model) showed improvements only up to the $2100^{\text{th}}$ training step, after which the test scores began to decline. This downturn indicates that the BT reward model or GenRM with SFT BoN response might be more susceptible to issues such as overfitting or reward hacking in later stages of training.

Accordingly, in our proposed approach, we increase the number of prompts allocated to RTV-supervised tasks and place an early emphasis on mathematical and coding tasks, supervised respectively by GenRM (with ground-truth data) and RTV. We anticipate that this strategy will enable the model to achieve optimal overall performance across various task types: those supervised by RTV, those supervised by GenRM with ground truth references, and those supervised by GenRM with SFT Best-of-N responses. This approach is expected to yield the best combined results, especially by allowing the model to reach peak performance on tasks in the last category before reward-hacking issues emerge.

**Early Acquisition of Fine-Grained Response Differences Enhances Performance Scaling.** Although the observed overall performance improvement can be partially explained by mitigating reward hacking issues associated with tasks supervised by GenRM with SFT Best-of-N responses, the specific performance boost in mathematical and coding tasks still merits further investigation. As illustrated in Figure 11, our 'Data Scale' method achieves substantially better performance on math and coding tasks compared to the initial run. Notably, this improvement occurs despite our method utilizing roughly the same number of prompts for mathematical and coding tasks as in the initial run.

Accordingly, we first analyze the types of prompts filtered by the Pre-PPO strategy. To conduct this analysis, we collect five responses per prompt, compute the maximum edit distance among these responses, and then categorize the prompts into separate bins based on these maximum edit distances. Next, we calculate the average normalized reward model score for each bin. In our view, the edit distance between responses can reflect the granularity of their differences to some extent—larger edit distances indicate coarser-grained differences, whereas smaller distances suggest finer-grained distinctions. As illustrated in Figure 12, we have the following observations and findings:

- Prompts supervised by GenRM with ground truth (e.g., mathematical and logical tasks) and those supervised by GenRM without ground truth (e.g., creative writing and cosplay tasks) exhibit distinctly different trends in normalized reward-model scores as the edit distance varies. These trends highlight fundamental differences in how the model learns across task types: **for tasks supervised by GenRM without ground truth, the model readily captures coarse-grained**

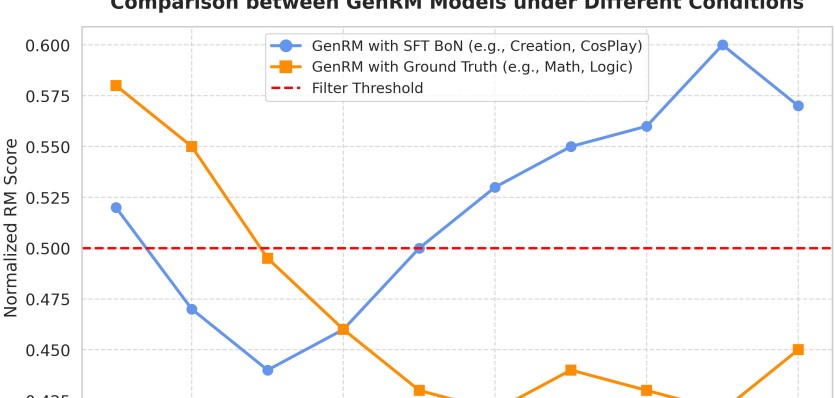

Figure 12: Comparison of Reward Model Scores across Different Edit Distance Bins for GenRM with and without Ground Truth.

**differences; whereas for tasks supervised by GenRM with ground truth, the model shows greater sensitivity to fine-grained distinctions.**

- In the Pre-PPO strategy, we explicitly exclude prompts that exhibit fine-grained response differences in mathematical and logical tasks, as well as those reflecting coarse-grained differences in creative writing and cosplay tasks. A subsequent ablation study suggests that reintroducing the previously excluded mathematical and logical prompts still delivers marginal improvements in overall performance. This finding implies that **learning coarse-grained patterns from creative writing and cosplay tasks negatively impacts the scalability of RLHF data**.

- We hypothesize that emphasizing mathematical and coding tasks during early training may also guide the model towards capturing fine-grained distinctions first, **thereby mitigating potential adverse effects from prematurely learning coarse-grained patterns.**

Furthermore, we analyze how reward score differences vary across prompt bins categorized by their maximum edit distances for different reward models. As shown in 13, both GenRM with ground truth and RTV assign larger score differences within bins corresponding to smaller edit distances. Conversely, GenRM without ground truth fails to produce meaningful score differences within these lower edit-distance bins. These results suggest that **reward models leveraging ground truth data or verification feedback demonstrate stronger sensitivity to fine-grained response variations compared to models trained without explicit ground truth supervision**. Moreover, when directly comparing RTV and GenRM equipped with ground truth data, RTV exhibits consistently larger score differences at low edit distances, highlighting RTV's enhanced capability in capturing subtle response distinctions.

**Deterioration of Model Response Diversity.** Finally, we compare the model response entropy between the baseline method and our proposed method. As illustrated in Figure 6 in the Appendix, the response entropy for creative writing and cosplay tasks is higher in our approach compared to the baseline. In contrast, our method achieves lower response entropy for mathematics, coding, and other reasoning tasks. These findings suggest that removing coarse-grained pattern prompts from tasks supervised by GenRM without ground truth alleviates the decline in model response diversity typically observed during RLHF. This strategy thus enables the model to better capture fine-grained differences in reasoning task responses, thereby enhancing the data-scaling effectiveness of RLHF.

## E  Discussions

**Q1: Do prompts with large edit distances negatively impact model performance, and should the model avoid learning from them?**

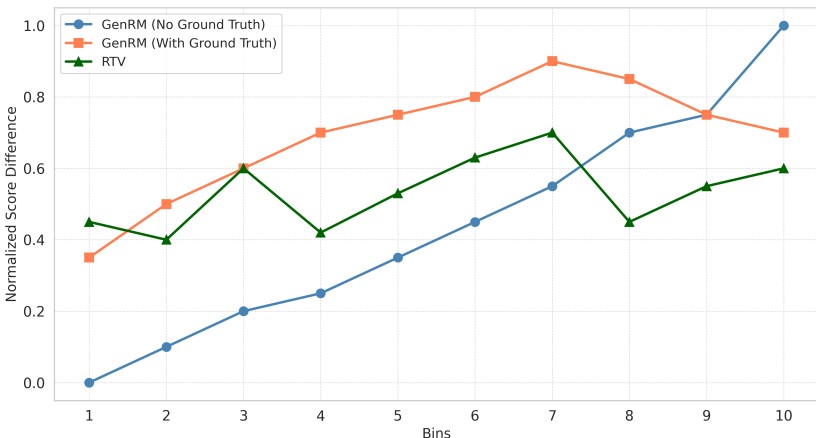

Figure 13: Comparison of score differences across varying edit-distance bins for GenRM (with and without ground truth) and RTV. Scores provided by GenRM without ground truth align with the response edit distance, indicating that larger edit distances—which represent greater response differences—correspond to larger score differences. However, RTV and GenRM with ground truth do not exhibit this trend. This suggests that GenRM effectively detects large differences between responses but struggles to identify smaller differences.

**A1:** Actually, prompts with larger edit distances (coarse-grained variations) and smaller edit distances (fine-grained variations) both contribute positively to improving the model. However, prioritizing coarse-grained (large edit distance) data early in training can adversely affect the model's ability to effectively learn fine-grained (small edit distance) distinctions later. Ideally, we want the model to thoroughly master fine-grained variations first before transitioning to learning from coarse-grained data.

**Q2: The "O1 series" method introduces long chain-of-thought (CoT) responses, which theoretically increases the edit distance of all response pairs. Why is this approach still effective?**

**A2:** The "O1 series" can essentially be viewed as transforming all fine-grained differences into sufficiently large, coarse-grained ones. By doing this, the model can more clearly categorize and generalize variations across different granularities. Personally, this approach seems like a comprehensive solution for handling varying levels of granularity. However, we acknowledge that there might exist deeper insights or interpretations beyond our current understanding.

**Q3: According to this analysis, should we first train our models on data with smaller edit distance variations (fine-grained responses), and later on data with larger edit distance variations (coarse-grained responses)?**

**A3:** We haven't conducted experiments as detailed as this due to several practical considerations. Edit distance is merely a coarse proxy for defining granularity levels, and its computational overhead is quite high. Hence, it's suitable for exploratory understanding rather than as a practical training strategy. Nevertheless, we have performed a similar strategy experiment in which tasks with abundant fine-grained variations (such as math and coding tasks) are trained first, followed by broader, coarsely-varied data later on. This approach demonstrated improved final performance. Importantly, this improvement depends heavily on the capacity of the verifier and GenRM with ground truth to accurately perceive fine-grained variations.

**Q4: What practical insights does this analysis provide us?**

**A4:** The concept of "fine-grained control" was initially highlighted by Anthropic when introducing Constitutional AI (CAI) [36]. The creation of CAI was inspired by the realization that methods such as Reinforcement Learning from AI Feedback (RLAIF) alone cannot directly capture human-preferred, fine-grained distinctions. To address this limitation, Anthropic proposed CAI, which explicitly encourages generative reward models (Gen-RM) to become sensitive to subtle aspects of prompts and responses. For example, CAI aims to help models detect subtle prompt-level linguistic nuances—such

as differences between requests for "critical reviews" versus just "reviews"—as well as recognize sophisticated vocabulary that enhances response quality, exemplified by Claude frequently using poetic phrases like "moonlight as in a dream" in literary creation tasks. Therefore, moving forward, we can build upon these insights by carefully constructing CAI-style humanistic datasets aimed specifically at training GenRMs without ground truth to perceive subtle, fine-grained distinctions. Subsequently, reinforcement learning (RL) techniques could then leverage these enhanced reward models, progressively improving their capability for fine-grained control over generated responses.

# F  Case Study

We collect the human feedback to compare the baseline and our model. However, due to the complexity of the prompts, we only list a shot case comparsion in our paper.

## F.1  Case 1 - STEM

We present a STEM case study demonstrating that the response of our model exhibits enhanced logical coherence (e.g. line 7-8 in the response of our model vs. line 12-14 in the response of baseline model ) and better informational density.

```
1  <-- PROMPT -->
2  It is known that the eye color of a certain type of fruit fly,
       white and black, is a pair of relative traits controlled by
        a pair of alleles on an autosome. Black is dominant over
       white, and the gene controlling black eyes is not expressed
        in female fruit flies. Therefore, male fruit flies have
       both black and white eyes, while female fruit flies only
       have white eyes. Additionally, there is a lethal homozygous
        dominance for the black eye gene. At the same time, short
       wings and long wings are controlled by another pair of
       alleles on an autosome, with long wings being dominant over
        short wings, and there is a lethal homozygous dominance
       for the long wing gene. A black-eyed, long-winged male is
       crossed with a white-eyed, long-winged female, and it is
       found that the ratio of black-eyed to white-eyed male fruit
        flies in the F1 generation is 1:1. All long-winged fruit
       flies from F1 are then allowed to freely mate. Please
       deduce the theoretically expected traits and ratios of the
       F2 male individuals.
3  The line aipc_data_dir.encode('utf-8') encodes a string into a
       UTF-8 formatted byte sequence. What are the benefits of
       this?
```

```
1  <-- RESPONSE OF OUR MODEL -->
2  1. **Determine the parental genotypes**:
3     - Let the gene controlling eye color be represented by \(A\)
        and \(a\), and the gene controlling wing length by \(B\)
        and \(b\).
4     - It is known that black eyes are dominant and homozygous
       dominant is lethal. Male fruit flies have both black and
       white eyes, while female fruit flies only have white eyes.
       A black-eyed male fruit fly has the genotype \(Aa\) (since
       \(AA\) is lethal), and a white-eyed female fruit fly could
       have the genotype \(Aa\) or \(aa\). The F1 generation shows
        a 1:1 ratio of black-eyed (\(Aa\)) to white-eyed (\(aa\))
       male fruit flies, indicating that the parental white-eyed
       female fruit fly has the genotype \(aa\).
5     - Since long wings are dominant over short wings and
       homozygous dominant for long wings is lethal, long-winged
```

individuals have the genotype \(Bb\). Therefore, the
parental black-eyed, long-winged male fruit fly has the
genotype \(AaBb\), and the white-eyed, long-winged female
fruit fly has the genotype \(aaBb\).

2. **Calculate the genotypes and ratios of the F1 long-winged
fruit flies**:
   - For the wing length trait, the parental cross is \(Bb \
times Bb\). According to the law of segregation, the
offspring genotypes and ratios are \(BB:Bb:bb = 1:2:1\).
Since \(BB\) is lethal, the F1 long-winged fruit flies have
the genotype \(Bb\), accounting for \(\frac{2}{3}\) of the
surviving individuals, and short-winged fruit flies have
the genotype \(bb\), accounting for \(\frac{1}{3}\) of the
surviving individuals. Thus, all F1 long-winged fruit flies
are \(Bb\).

3. **Calculate the genotypes and ratios of the F2 generation
produced by free mating of F1**:
   - F1 long-winged fruit flies \(Bb\) freely mate. According
to the law of segregation, the gene frequency of \(B\) is
\(\frac{1}{2}\), and the gene frequency of \(b\) is \(\frac
{1}{2}\). Using the Hardy-Weinberg law (\((p+q)^2 = p^2 + 2
pq + q^2\), where \(p\) and \(q\) are the frequencies of
the alleles), the theoretical offspring genotypes and
ratios are \(BB:Bb:bb=1:2:1\). However, since \(BB\) is
lethal, the actual ratios are \(Bb:bb = 2:1\), meaning long
-winged fruit flies (\(Bb\)) account for \(\frac{2}{3}\)
and short-winged fruit flies (\(bb\)) account for \(\frac
{1}{3}\).
   - For eye color, the parental cross is \(Aa \times aa\), and
the F1 genotypes and ratios are \(Aa:aa = 1:1\). F1
individuals freely mate, with the gene frequency of \(A\)
being \(\frac{1}{4}\) and the gene frequency of \(a\) being
\(\frac{3}{4}\). The theoretical offspring genotypes and
ratios are \(AA:Aa:aa = 1:6:9\). Since \(AA\) is lethal,
the actual ratios are \(Aa:aa = 6:9 = 2:3\), meaning \(Aa\)
accounts for \(\frac{2}{5}\) and \(aa\) accounts for \(\
frac{3}{5}\).

4. **Calculate the ratios of different traits in F2 male
individuals**:
   - For the two pairs of relative traits, wing length and eye
color, the traits of F2 male individuals are calculated by
combining the two according to the law of independent
assortment.
   - Black-eyed, long-winged (\(AaBb\)): \(\frac{2}{5} \times \
frac{2}{3} = \frac{4}{15}\)
   - Black-eyed, short-winged (\(Aabb\)): \(\frac{2}{5} \times
\frac{1}{3} = \frac{2}{15}\)
   - White-eyed, long-winged (\(aaBb\)): \(\frac{3}{5} \times \
frac{2}{3} = \frac{6}{15}\)
   - White-eyed, short-winged (\(aabb\)): \(\frac{3}{5} \times
\frac{1}{3} = \frac{3}{15}\)
   - Therefore, the traits and ratios of F2 male individuals
are black-eyed, long-winged : black-eyed, short-winged :
white-eyed, long-winged : white-eyed, short-winged \( =
4:2:6:3\).

22 | In conclusion, the theoretically expected traits and ratios of F2 male individuals are black-eyed, long-winged : black-eyed, short-winged : white-eyed, long-winged : white-eyed, short-winged $( = 4:2:6:3)$.

---

1 | <-- RESPONSE OF BASELINE MODEL -->
2 | This is a question about the laws of genetic inheritance and the derivation of trait ratios. The key to solving the problem lies in clarifying the dominant-recessive relationships of the genes, the lethality conditions, and deducing the parental genotypes based on the given mating combinations and offspring ratios. Then, using the method of calculating free mating, the trait ratios of the offspring can be determined.
3 |
4 | 1. **Determining the parental genotypes for eye color**:
5 | - It is given that black eyes (denoted as $A$) are dominant over white eyes (denoted as $a$), and the gene controlling black eyes is not expressed in female fruit flies. Male fruit flies have either black or white eyes, with the homozygous dominant genotype for black eyes ($AA$) being lethal.
6 | - A black-eyed male ($A\_$) is crossed with a white-eyed female. The $F1$ generation of male fruit flies shows a black-eyed ($A\_$) to white-eyed ($aa$) ratio of $1:1$. Based on the test cross results, it can be deduced that the black-eyed male parent has the genotype $Aa$ for eye color, while the white-eyed female parent has the genotype $aa$.
7 |
8 | 2. **Determining the parental genotypes for wing type**:
9 | - It is given that long wings (denoted as $B$) are dominant over short wings (denoted as $b$), and the homozygous dominant genotype for long wings ($BB$) is lethal.
10 | - The parents are a long-winged male ($B\_$) and a long-winged female ($B\_$). Since the homozygous dominant genotype for long wings is lethal, both parents must have the genotype $Bb$ for wing type.
11 |
12 | 3. **Deriving the complete parental genotypes and the $F1$ long-winged fruit fly genotype ratios**:
13 | - Combining the above analysis, the parental genotypes are $AaBb$ for the male and $aaBb$ for the female.
14 | - For wing type, the $F1$ long-winged fruit flies have genotypes and ratios of $Bb:bb = 2:1$, meaning $Bb$ accounts for $\frac{2}{3}$ and $bb$ accounts for $\frac{1}{3}$.
15 |
16 | 4. **Calculating the trait ratios for wing type in the $F2$ generation**:
17 | - The $F1$ long-winged fruit flies mate freely. The frequency of the $B$ allele in the gametes produced by $F1$ is $\frac{2}{3} \times \frac{1}{2} = \frac{1}{3}$, and the frequency of the $b$ allele is $1 - \frac{1}{3} = \frac{2}{3}$.
18 | - According to the Hardy-Weinberg law, the genotype frequencies in $F2$ are: $BB = (\frac{1}{3})^2 = \frac{1}{9}$ (lethal), $Bb = 2 \times \frac{1}{3} \times \frac{}{}$

```
    {2}{3} = \frac{4}{9}\), and \(bb = (\frac{2}{3})^2 = \frac
    {4}{9}\). Thus, the ratio of long-winged (\(Bb\)) to short-
    winged (\(bb\)) individuals is \(\frac{4}{9}:\frac{4}{9} =
    1:1\).
19
20 5. **Calculating the trait ratios for eye color in the \(F2\)
    generation**:
21   - The \(F1\) genotypes for eye color and their ratios are \(
    Aa:aa = 1:1\). The frequency of the \(A\) allele in the
    gametes produced is \(\frac{1}{2} \times \frac{1}{2} = \
    frac{1}{4}\), and the frequency of the \(a\) allele is \(1
    - \frac{1}{4} = \frac{3}{4}\).
22   - The genotype frequencies in \(F2\) are: \(AA = (\frac
    {1}{4})^2 = \frac{1}{16}\) (lethal), \(Aa = 2 \times \frac
    {1}{4} \times \frac{3}{4} = \frac{6}{16}\), and \(aa = (\
    frac{3}{4})^2 = \frac{9}{16}\). Thus, the ratio of black-
    eyed (\(Aa\)) to white-eyed (\(aa\)) individuals is \(\frac
    {6}{16}:\frac{9}{16} = 2:3\).
23
24 6. **Calculating the trait ratios for male individuals in the
    \(F2\) generation**:
25   - Since the inheritance of eye color and wing type are
    independent events, the ratio of black-eyed, long-winged
    males in \(F2\) is \(\frac{2}{5} \times \frac{1}{2} = \frac
    {1}{5}\), black-eyed, short-winged males is \(\frac{2}{5} \
    times \frac{1}{2} = \frac{1}{5}\), white-eyed, long-winged
    males is \(\frac{3}{5} \times \frac{1}{2} = \frac{3}{10}\),
     and white-eyed, short-winged males is \(\frac{3}{5} \times
     \frac{1}{2} = \frac{3}{10}\).
26   - Therefore, the traits and ratios of \(F2\) male
    individuals are: black-eyed long-winged : black-eyed short-
    winged : white-eyed long-winged : white-eyed short-winged =
     \(2:2:3:3\).
27
28 In conclusion, the theoretical traits and ratios of \(F2\) male
     individuals are black-eyed long-winged : black-eyed short-
    winged : white-eyed long-winged : white-eyed short-winged =
     \(2:2:3:3\).
```

## F.2 Case 2 - Writing

We present a case study demonstrating how our model's responses exhibit superior linguistic elegance (e.g. title of essay) while meticulously adhering to nuanced instructions in prompts.

```
1 <-- PROMPT -->
2 You need to complete the article creation according to the
    following requirements:
3 1: The content of the article requires a rigorous structure and
     clear logic, with a strict argumentation process from
    raising the issue, analyzing the problem, to solving it.
4 2: The title of the article needs to be in a professional style
    , succinctly listing the main information of the event in a
     straightforward news-reporting manner. Amounts of money
    involved should be reflected in the title.
5 3: Please strictly follow the requirements for output, the
    content must comply with the law, and it is prohibited to
    evaluate the law and government law enforcement departments
    . Adhere to the WeChat public account community norms,
    prohibit the appearance of violent, pornographic, and
```

sensitive vocabulary, and prohibit the output of any content unrelated to the article. It is forbidden to deviate from the core focus of the event, keeping the article content closely tied to hot events.

4: The article layout requires that the title and content, the first line outputs the title of the article (do not start with the word "title"), and the use of book title marks is prohibited. The second line starts with the specific content of the article. The full text should be around 2000 words. The article content must have subheadings, and the subheadings should be expressed in a straightforward manner, with the use of book title marks prohibited in subheadings.

The theme is: "Urban Waste Sorting", this theme cannot be used as the title, write an article of about 2000 words.

---

<-- RESPONSE OF OUR MODEL -->

Urban Waste Sorting: Current Status, Challenges, and Breakthrough Strategies

Exploring New Paths for Sustainable Development

**Significant Achievements in Waste Sorting Implementation**

In recent years, with the gradual increase in environmental awareness, urban waste sorting has become a focal topic of social concern. Many cities have actively responded to national calls, vigorously promoting waste sorting initiatives, and have achieved remarkable results in improving environmental quality and enhancing resource utilization rates.

Taking Shanghai as an example, since the official implementation of the waste sorting policy on July 1, 2019, the government has invested approximately 20 billion yuan in infrastructure construction and public education campaigns. Today, Shanghai's waste sorting efforts have yielded impressive outcomes. Neatly arranged sorting bins have become a common sight on the streets, while scenes of residents conscientiously sorting and disposing of waste are widespread in communities. According to statistics, the accuracy rate of waste sorting in Shanghai has significantly improved. The sorting and processing systems for dry, wet, recyclable, and hazardous waste are increasingly refined, with the resource utilization rate of waste exceeding 35\%. This has effectively reduced environmental pollution and laid a solid foundation for the city's sustainable development.

Beijing is not far behind. To advance waste sorting, the government has cumulatively invested around 15 billion yuan in building waste treatment facilities, purchasing sorting transport vehicles, and conducting promotional activities. Through a series of measures, Beijing's waste sorting efforts have achieved phased success. More residents are becoming aware of and participating in waste sorting, leading to a noticeable improvement in the city's environmental hygiene. Data shows that the recycling rate of domestic waste in Beijing has reached approximately 30\%, making a positive contribution to resource recycling.

**Challenges Facing Waste Sorting**

Despite notable achievements in some cities, it is undeniable that the nationwide promotion of urban waste sorting still faces numerous severe challenges.

**Residents' Awareness and Habits Need Improvement**

Some residents lack sufficient understanding of waste sorting and have inadequate environmental awareness and a sense of responsibility. Many find the process cumbersome, believing it increases their living and time costs, and are therefore reluctant to participate actively. Additionally, long-established habits of improper waste disposal are difficult to change quickly, resulting in low accuracy rates in waste sorting. For instance, in some older neighborhoods, indiscriminate waste disposal remains common, and recyclables are often mixed with other types of waste.

**Complex and Confusing Sorting Standards**

Currently, waste sorting standards in China are not fully unified, with variations across different regions. This creates significant confusion for residents and workers. For example, some cities classify waste into four categories, while others use five or more. The lack of uniformity makes it difficult for residents to grasp the correct sorting methods. Moreover, the specifics of waste sorting are often complex, with ambiguous boundaries between categories. For instance, whether large bones belong to kitchen waste or other waste is a common point of confusion for many residents.

**Lagging Infrastructure Development**

The development of waste sorting infrastructure lags behind actual needs. On one hand, waste collection containers are often inadequately placed, insufficient in number, and poorly labeled. In some streets and communities, uneven distribution of bins makes waste disposal inconvenient for residents. On the other hand, transportation and processing facilities are also lacking. In some cities, waste transport vehicles do not support sorted collection, leading to the remixing of different waste types during transportation, rendering previous efforts futile. Furthermore, waste processing capacity, especially for hazardous and kitchen waste, remains limited, with technological and operational capabilities needing improvement.

**Lack of Effective Supervision Mechanisms**

Supervision mechanisms for waste sorting are still underdeveloped, lacking effective regulatory measures and enforcement tools. Currently, most cities rely on community workers and volunteers for oversight, but limited manpower makes it difficult to monitor the entire waste sorting process comprehensively. Without clear penalties and strong enforcement for non-compliance, some residents do not take waste sorting regulations seriously, leading to frequent violations.

**Seeking Breakthrough Solutions Through Collaborative Efforts**

To further advance urban waste sorting and achieve sustainable development goals, it is essential for the government, businesses, and residents to work together and take effective measures to address current challenges.

**Strengthening Public Education**

The government should intensify efforts to educate the public about waste sorting, using various channels and methods to disseminate knowledge and raise environmental awareness and responsibility. For example, diverse promotional campaigns can be conducted through television, radio, newspapers, and online platforms. Workshops and training sessions can be organized in communities, schools, and workplaces. Engaging and informative materials, such as brochures and public service announcements, can be distributed and broadcasted. Additionally, themed activities like waste sorting competitions and family events can encourage participation and help residents develop good sorting habits.

**Unifying Sorting Standards**

The government should expedite the unification of waste sorting standards, creating a scientific, reasonable, and easy-to-understand classification guide. Clear definitions and boundaries for each waste category should be established to reduce ambiguity. Simultaneously, efforts should be made to promote and train residents and workers on these standards to ensure accurate implementation. Drawing on international best practices, China can refine its waste sorting standards and systems to better suit local conditions.

**Improving Infrastructure Development**

Increased investment in waste sorting infrastructure is crucial. Governments should plan and deploy waste collection containers more rationally, ensuring even distribution and clear labeling. Upgrading waste transportation and processing facilities is also essential to enable sorted collection and treatment. Advanced transport vehicles equipped with sorting mechanisms should be procured, and modern waste treatment plants should be built to enhance processing capacity and technological capabilities. Encouraging private sector participation in infrastructure development can also diversify funding sources.

**Establishing Robust Supervision Mechanisms**

A comprehensive supervision mechanism for waste sorting must be established to oversee the entire process. Governments should formulate clear regulatory measures and penalty standards, enforcing stricter consequences for violations. Leveraging the efforts of community workers, volunteers, and residents, a reporting and reward system can be implemented to encourage public participation in oversight. Additionally, technological solutions like surveillance cameras and smart bins can enable real-time monitoring and management.

**Promoting Industry Development**

The development of waste sorting-related industries should be actively promoted to form a complete industrial chain. Governments can introduce policies to encourage technological innovation and efficiency improvements in waste sorting. Support for waste recycling projects can enhance resource utilization. Furthermore, establishing industrial parks for waste sorting can cluster related businesses, fostering scalable and specialized development.

Urban waste sorting is a long-term and challenging task, integral to sustainable urban development and residents' quality of life. Despite current difficulties, with determination and collaborative efforts, effective measures can be taken to overcome these challenges and achieve comprehensive progress in waste sorting. Let us work together to build a beautiful, green, and sustainable future for our cities.

---

<-- RESPONSE OF BASELINE MODEL -->

"Challenges and Solutions in Urban Waste Sorting: A Multidimensional Analysis"

Current Status and Urgency of Urban Waste Sorting

With the acceleration of urbanization, the amount of waste generated in cities has grown significantly. Statistics show that the total annual waste output in major cities has continued to rise in recent years, making waste management a critical issue for urban development. Traditional waste disposal methods, such as landfilling and incineration, not only consume large amounts of land resources but also pose environmental risks. Waste sorting, as the front-end process of waste management, plays a key role in achieving waste reduction, resource recovery, and harmless disposal. It can effectively improve recycling rates, minimize environmental damage, and lay the foundation for sustainable urban development. However, although waste sorting initiatives have been gradually implemented in many Chinese cities, the overall results remain uneven, with numerous challenges persisting. The urgency of advancing waste sorting efforts is undeniable.

Challenges in Implementing Waste Sorting

Lack of Public Awareness and Habits

A considerable portion of urban residents lack sufficient knowledge about waste sorting. Many are unsure how to properly categorize different types of waste---such as distinguishing between hazardous, recyclable, and other waste. Even among those who understand the basics, deeply ingrained habits of mixed waste disposal are difficult to change in the short term. Some residents find sorting cumbersome and lack motivation to participate, leading to careless disposal. These issues in awareness and behavior make it difficult to enforce effective waste sorting at the source.

Inadequate Infrastructure

In some cities, waste sorting facilities are insufficient. Trash bins are not set up according to strict sorting

standards, or their labels are unclear, making proper disposal confusing for residents. Additionally, the ``sort-then-mix'' phenomenon occurs during waste collection and transportation---some areas have sorted bins, but collection trucks mix all waste together, rendering residents' efforts meaningless and discouraging participation. Furthermore, terminal waste treatment facilities face shortcomings, such as underdeveloped recycling systems and a lack of specialized hazardous waste disposal methods, preventing sorted waste from being properly processed.

Weak Enforcement and Supervision of Regulations

Although many regions have introduced waste sorting laws and regulations, enforcement remains inconsistent. Penalties for non-compliance are often too lenient to serve as an effective deterrent. Due to the vast urban areas and numerous waste generation points, regulatory bodies struggle to conduct comprehensive, real-time oversight, allowing violations to go unchecked. Moreover, insufficient public awareness of these regulations weakens their guiding and binding effects.

Strategies for Multi-Stakeholder Collaboration in Waste Sorting

Enhancing Public Education and Awareness

Governments and communities should intensify awareness campaigns through diverse activities such as community lectures, themed exhibitions, and online quizzes to educate residents on proper waste sorting. Tailored approaches should be used for different demographics---for example, integrating waste sorting into school curricula to instill environmental consciousness in youth, while using simple, accessible materials for elderly residents. Media platforms, including TV, radio, social media, and short videos, should also be leveraged to disseminate information and foster a culture of participation.

Improving Infrastructure

At the disposal stage, clearly labeled and strategically placed sorting bins should be provided to facilitate correct waste separation. Regular maintenance and upgrades of these facilities are essential. For collection and transportation, dedicated vehicles for different waste categories must be deployed to prevent mixed collection. An efficient waste transfer system should also be established. On the processing end, investments should be made in recycling and hazardous waste treatment facilities, incorporating advanced technologies to enhance disposal efficiency.

Strengthening Regulation and Enforcement

Regulatory bodies must enhance daily supervision and enforcement, establishing performance-based evaluation systems to reward success and address shortcomings. Violations should be penalized strictly to increase accountability. Public reporting mechanisms can also encourage residents to monitor and report non-compliance, fostering collective oversight. Additionally, continuous

```
         legal education should ensure residents understand and
         adhere to waste sorting requirements.
26
27  Innovative Models for Sustainable Waste Sorting
28  Introducing Market Mechanisms
29  Businesses should be encouraged to participate in the waste
         sorting industry chain through policy incentives and
         subsidies, attracting private investment in collection,
         transportation, and processing. For instance, specialized
         waste sorting service providers can offer comprehensive
         solutions to communities and institutions. A recycling
         industry system can also be developed to monetize
         recyclables, creating a circular economy. Innovative
         business models, such as reward points or exchange programs
         , can further motivate public participation.
30
31  Leveraging Technology
32  Smart waste sorting technologies---such as AI-powered bins and
         recycling machines using facial recognition or QR codes---
         can track and incentivize proper sorting through reward
         systems. Big data and IoT can enable real-time monitoring
         and analysis of waste generation and sorting rates,
         supporting data-driven decision-making. Additionally,
         advancements like AI-based sorting robots can improve
         efficiency in waste processing.
33
34  Conclusion
35  Urban waste sorting is a systematic endeavor requiring changes
         in public habits, infrastructure upgrades, regulatory
         enforcement, and societal collaboration. Only through
         robust education, improved infrastructure, strict oversight
         , and continuous innovation can cities effectively advance
         waste sorting, enhance environmental quality, and achieve
         sustainable development---creating a better living
         environment for all residents.
```

### F.3 Case 3 - Coding Knowledge

We present a case study demonstrating our model's enhanced response quality, characterized by better information accuracy and content richness, as exemplified by the provided code sample (line 12-23).

```
1  <-- PROMPT -->
2  The line aipc_data_dir.encode('utf-8') encodes a string into a
         UTF-8 formatted byte sequence. What are the benefits of
         this?
```

```
1  <-- RESPONSE OF OUR MODEL -->
2  **Encoding strings into UTF-8 byte sequences offers several key
          benefits in Python and broader programming contexts:**
3
4  1. **Broad Compatibility**: UTF-8 is a widely accepted and
         supported character encoding standard. It is the most
         common encoding on the internet, with nearly all modern
         operating systems, programming languages, and applications
         supporting it. This ensures reliable data transmission and
         processing across different systems, avoiding garbled text
         due to encoding incompatibilities. For example, when
         transmitting text data in web applications, UTF-8 is the
```

standard encoding, correctly interpreted by browsers,
    servers, and other components.

2. **Variable-Length Encoding**: UTF-8 uses a variable-length
   encoding scheme, representing characters with 1 to 4 bytes.
    For ASCII characters (basic Latin letters, numbers, and
   symbols), UTF-8 uses a single byte, maintaining
   compatibility with ASCII. For non-ASCII characters, it uses
    2 to 4 bytes as needed. This efficiency makes UTF-8 ideal
   for storage and transmission, especially when the data
   contains many ASCII characters.

3. **Multilingual Support**: UTF-8 can represent nearly all
   global writing systems, including letters, symbols, Chinese
   , Japanese, Korean, and more. This makes it perfect for
   internationalized and multilingual applications. Regardless
    of the language, UTF-8 provides a unified encoding
   solution.

4. **Backward Compatibility**: Since UTF-8 encodes ASCII
   characters identically to ASCII, it maintains backward
   compatibility with legacy ASCII-based systems. This allows
   seamless upgrades to support broader character sets without
    disrupting existing infrastructure.

### Example Code:
```python
# Define a string containing Chinese characters
aipc_data_dir = "hello, world"
# Encode the string into UTF-8 byte sequence
encoded_data = aipc_data_dir.encode('utf-8')
print(encoded_data)

# Decode the UTF-8 byte sequence back to a string
decoded_data = encoded_data.decode('utf-8')
print(decoded_data)
```
This example demonstrates encoding a Chinese string into UTF-8
    bytes and decoding it back, showcasing UTF-8's reliability
    in handling multilingual text.

In summary, UTF-8 encoding is a robust and universal choice,
    ensuring smooth data exchange across systems. Whether for
    web development, text processing, or system
    interoperability, UTF-8 remains a trusted standard.

---

<-- RESPONSE OF BASELINE MODEL -->
The primary benefits of encoding strings into UTF-8 byte
    sequences are as follows:

1. Universality and Compatibility
Cross-platform Support: UTF-8 is a universal character encoding
     standard supported by nearly all modern operating systems,
     programming languages, and network protocols. Data encoded
     in UTF-8 can be properly processed across Windows, Linux,
    and macOS systems. For example, a text file stored with UTF
    -8 encoding on a Linux server can be read on a Windows
    client without garbled characters, as Windows systems
    equally support UTF-8 encoding.

Multilingual Capability: UTF-8 can represent characters from
    nearly all writing systems worldwide, covering languages
    from different countries and regions. Whether it's English,
     Chinese, Arabic, Japanese, or others, UTF-8 can encode
    them all. This makes UTF-8 highly practical for handling
    multilingual content, such as in internationalized websites
     or software applications. For instance, a global e-
    commerce platform dealing with product descriptions and
    user comments in various languages can rely on UTF-8
    encoding to ensure proper storage and display of all text.

2. Network Transmission
Efficiency: UTF-8 offers efficiency advantages in network
    transmission. For ASCII characters (basic English letters,
    numbers, and common symbols), UTF-8 encoding uses only 1
    byte, identical to traditional ASCII encoding, ensuring no
    additional overhead when transmitting English text. For non
    -ASCII characters, UTF-8 uses 2 to 4 bytes depending on the
     character. Compared to fixed-length encodings like UTF-16
    (which typically uses 2 or 4 bytes for all characters), UTF
    -8 may occupy less space when transmitting text containing
    predominantly ASCII characters, thereby improving network
    efficiency. For example, when transmitting an English news
    article with minimal Chinese content, UTF-8 encoded text
    consumes less bandwidth than UTF-16 encoded text.

Protocol Support: Many network protocols, such as HTTP, SMTP,
    and POP3, default to or recommend UTF-8 encoding. When
    transmitting web content via HTTP, using UTF-8 ensures that
     all characters (including text in different languages and
    special symbols) are correctly parsed and displayed by
    browsers.

3. Data Storage
Space Efficiency: As mentioned earlier, UTF-8 uses 1 byte for
    ASCII characters and typically 3 bytes for common Chinese
    characters. Compared to fixed-length encodings like UCS-4 (
    which uses 4 bytes per character), UTF-8 can significantly
    reduce storage space for text containing large amounts of
    ASCII or commonly used characters. This is particularly
    important for databases and file systems storing
    substantial text data. For example, in a database table
    storing user comments, UTF-8 encoding can dramatically
    reduce storage requirements and lower costs compared to UCS
    -4 encoding.

Flexibility: UTF-8's variable-length encoding provides great
    flexibility in storing different types of characters. It
    dynamically allocates bytes based on actual character
    requirements, meeting the need to represent all characters
    while minimizing space usage for common ones.

