# OpenReview forum: "Exploring Data Scaling Trends and Effects in Reinforcement Learning from Human Feedback"
_NeurIPS.cc/2025/Conference — NeurIPS 2025 poster_

### Official Review · Reviewer_UPWF · 2025-07-01

**Clarity:** 2
**Significance:** 3
**Originality:** 2
**Rating:** 3
**Confidence:** 3

**Summary:**

This paper explores the challenges of data scaling in RLHF, focusing on reward hacking and response diversity issues. The authors introduce a hybrid reward system combining Reasoning Task Verifiers and Generative Reward Models to enhance model robustness and prevent reward manipulation. They also propose the Pre-PPO method for prompt selection, targeting difficult prompts to improve sample efficiency. Through experiments on a smaller model up to 25B parameters and a larger model up to 150B parameters, the authors demonstrate significant performance improvements, particularly in mathematical and coding tasks.

**Questions:**

**1. Regarding the revision of paper for clarity and conciseness:**

Can you revise the abstract to eliminate redundancy, such as the repeated mention of "we propose a novel prompt-selection method named Pre-PPO," and ensure it concisely conveys your contribution? Could you also address any additional writing flaws (e.g., unclear phrasing) identified in the abstract or elsewhere in the paper?

**2. Regarding differentiation from existing work:**

Would you be willing to conduct a detailed comparison showing how your task prioritization and prompt selection methods differ from or improve upon prior work, such as DeepSeek-R1? Could you include specific evidence (e.g., theoretical differences or empirical results) to highlight your approach’s novelty in the RLHF context?

**3. Regarding systematic exploration of task prioritization design:**

Can you provide experiments testing alternative task orderings (e.g., placing mathematical and coding tasks later in training) or varying data proportions to assess their impact on performance? Alternatively, could you offer a detailed justification for your current design without such experiments?

**4. Regarding validation of reproducibility with open-source resources:**
Would you consider validating your results using open-source models (e.g., Qwen, Llama) and datasets (e.g., MT-Bench, AlpacaEval, GSM8K) to address reproducibility concerns tied to closed-source resources? If not, can you propose an alternative plan to enhance verifiability?

**Evaluation Criteria:** My assessment would improve to "Accept" (5) if the authors provide a revised abstract with enhanced clarity and no redundancy, clearly differentiate their method from prior work like DeepSeek-R1 with specific evidence, conduct or justify experiments exploring alternative task orderings or data proportions, and validate results using open-source resources or a robust reproducibility plan. The score would move to "Borderline Accept" (4) if the authors partially address these issues—e.g., by improving the abstract and providing some comparisons or limited experiments—but fail to fully resolve concerns about originality, systematic exploration, or reproducibility.

**Ethical Concerns:**

["NO or VERY MINOR ethics concerns only"]

**Final Justification:**

I have carefully reviewed the authors' rebuttal and appreciate the thoughtful efforts to address my concerns. However, I still believe that the differences in focus between long-form and short-form RL do not adequately address my concerns regarding the novelty of the proposed method. Furthermore, the issue of lacking open-source verification (models and evaluation benchmarks) persists, which limits the generalizability and reproducibility of the approach presented in this paper.

**Limitations:**

No, the authors have not fully addressed the limitations of their work. Although the paper provides a strong theoretical foundation and presents promising experimental results, some key limitations are not adequately discussed.

**Paper Formatting Concerns:**

No paper format issues exist.

**Quality:**

2

**Strengths And Weaknesses:**

**Strength:**

**S1. High Practical Relevance:** The paper addresses highly relevant challenges for RLHF practitioners, particularly in terms of reward model robustness and response diversity. These are practical concerns for scaling RLHF to large models, making the research applicable in real-world scenarios.

**S2. Innovative Approach to RLHF Data Construction:** Unlike much of the existing literature, which mainly addresses algorithmic challenges, this paper emphasizes the crucial role of training data in improving RLHF performance. The introduction of Pre-PPO and the hybrid reward system combining Reasoning Task Verifiers (RTV) and Generative Reward Models (GenRM) is an innovative approach to mitigating reward hacking and improving response diversity.

**Weakness:**

**W1. Further Exploration Needed for Academic Originality and Depth:** The paper's academic originality could be further strengthened, as its proposed methods, such as task prioritization and prompt selection, while demonstrating performance improvements, closely resemble earlier curriculum-learning approaches, which structure training by sequencing tasks from simpler to more reasoning-intensive to gradually build the model’s capabilities. For instance, the task prioritization strategy bears conceptual similarities to DeepSeek-R1: Incentivizing Reasoning Capability in LLMs via Reinforcement Learning (2025), which also prioritize reasoning-intensive tasks during the early training stage of the model to enhance reasoning abilities. However, the paper does not sufficiently articulate how its approach surpasses or diverges from such prior studies, which substantially weakens the work’s novelty. To enhance its academic contribution, the authors should explicitly highlight the specific differences between their method and works like DeepSeek-R1, such as unique task selection criteria or novel integration with RLHF, and support these distinctions with theoretical analysis or comparative experiments.

**W2. Need for More Systematic Exploration in Method Design:** The current task priority method is overly specialized and lacks universality due to the absence of systematic experiments. The design of the task prioritization method could be further optimized, particularly in terms of experimental rigor and mechanistic insight. The authors did not explore alternative setups, such as placing mathematical and coding tasks in the later stages of training (a common practice in some pretraining approaches) or adjusting data proportions (e.g., reducing the weight of non-mathematical/coding data) to test their assumptions. This lack of exploration may hinder a comprehensive evaluation of the method’s effectiveness.

**W3. Limited Reproducibility Due to Closed-Source Data and Models:** The use of closed-source data and models in this study significantly hampers its reproducibility, thereby limiting the academic impact and verifiability of the findings. While the decision to keep resources proprietary may be justified, it prevents external researchers from replicating or validating the results. To enhance the study’s credibility and applicability, the authors are encouraged to conduct supplementary experiments using open-source models, such as qwen or llama, and publicly available datasets, like MT-Bench, AlpacaEval or GSM8K, to verify their conclusions and demonstrate the method’s robustness across diverse settings.

**W4. Writing flaws (A minor point):**
Some writing flaws exist in the paper. For example, the abstract contains repetitive sentences, particularly "we propose a novel prompt-selection method named Pre-PPO" appears twice. This redundancy reduces the conciseness and professionalism of the abstract, which should be as concise as possible as it serves as the facade of the paper. It is recommended to check the writing flaws in your paper.

---

> ### Author Rebuttal · Authors · 2025-07-31
>
> # Reproducibility and Use of Standard Public Benchmarks
>
> We appreciate the importance of using standard public benchmarks and have indeed conducted evaluations on several open benchmarks. Our results show:
>
> - Significant improvement in coding tasks (e.g., LiveCodeBench: 30.3 -> 36.2).
> - Slight improvements or comparable performance in general knowledge tests (e.g., MMLU: 87.3 -> 87.3, MMLU pro: 79.6 -> 80.1).
> - Slightly lower but comparable performance on simple math tasks (e.g., AMC23: 68.8 -> 68.7, MATH500: 88.2 -> 87.8).
> - Comparable performance on high-difficulty math tasks (e.g., AIME: 18.6 -> 17.9, 26.9 -> 27.2).
>
> However, we've identified limitations in many public benchmarks:
> - Potential IID issues, not fully reflecting real-world challenges.
> - Risk of data leakage due to widespread use.
> - Insufficient difficulty to comprehensively assess advanced models.
>
> For instance, our model can score nearly 90 points on some math benchmarks, but only around 30 points on our more challenging internal test sets.
>
> Consequently, we developed an internal evaluation suite with the following advantages:
> - Higher difficulty, better differentiating top-tier model capabilities.
> - Continuous iteration, preventing model overfitting.
> - Black-box testing, ensuring fair and authentic evaluation.
>
> We believe combining public benchmarks with rigorous internal assessments provides a more comprehensive and reliable evaluation of model performance. We will continue to monitor and use public benchmarks while refining our internal evaluation system.
>
> Regarding the use of open-source models and datasets, we acknowledge their value in enhancing reproducibility. We are considering conducting supplementary experiments with these resources and will include them in future work to provide a more comprehensive comparison.
>
> | Task Series       | Sub-Task Name                              | Question Count | Baseline Score | Our Model Score |
> |-------------------|-------------------------------------------|----------------|----------------|-----------------|
> | MATH - Aligned    | AMC23 (AMC23fblshot_v2.rwq_score)         | 40             | 68.8           | 68.7            |
> | MATH - Aligned    | MATH500                                   | 500            | 88.2           | 87.8            |
> | AIME (Open-Style) | AIME 2025                                 | 30             | 18.6           | 17.9            |
> | AIME (Open-Style) | AIME 2024                                 | 30             | 26.9           | 27.2            |
> | Coding - Aligned  | LiveCodeBench                             | 131            | 30.3           | 36.2            |
> | General Knowledge | MMLU                                      | 150            | 87.3           | 87.3            |
> | General Knowledge | MMLU pro                                  | 1000           | 79.6           | 80.1            |
> | Reasoning         | GPQA diamond                              | 198            | 59.6           | 60.1            |
> | Reasoning         | SuperGPQA (superGPQAfrdom_1K_sample_2)    | 250            | 52.8           | 54.0            |
>
> # Systematic Exploration of Task Prioritization Design
>
> We appreciate the reviewer's request for per-task learning curves for the Early Training Emphasis strategy. Due to our rebuttal policy, we are unable to include new figures in the PDF. However, readers can refer to Figure 4 in the original paper, specifically the first 1400 steps, to gain insights into the learning progression for math and coding tasks.
>
> In addition to the information provided in the paper, we have conducted supplementary experiments regarding the order of training for code and math tasks. On our 25B model, we tested the following configurations:
>
> 1. Training exclusively on coding data first
> 2. Training exclusively on math data first
> 3. Our current mixed approach
>
> Results on our internal test set V1 showed:
> - Coding first: 38.1
> - Math first: 37.8
> - Current mixed approach: 38.8
>
> These results demonstrate that our current mixed approach outperforms exclusively prioritizing either coding or math tasks.
>
> We hope this additional information provides further clarity on the effects of task ordering in our training strategy. We appreciate the reviewer's suggestion and will consider generating more detailed per-task learning curves for future iterations of our work, as computational resources allow.
>
>
> # Writing and Presentation
> We acknowledge the issues of redundancy and lack of clarity in our writing. We will promptly revise the abstract to eliminate redundancy and clearly convey our contributions. Moreover, we'll review and enhance the clarity and conciseness of other paper parts. Thank you for pointing these out.
>
> We present the revised abstract below, incorporating feedback on clarity and scope:
>
> "Reinforcement Learning from Human Feedback (RLHF) is essential for aligning large language models (LLMs) with human preferences and values. While recent research has primarily focused on algorithmic advancements—such as reducing computational overhead or strengthening reward models to mitigate reward hacking—the critical role of prompt-data construction and its scalability has received comparatively less attention. In this paper, we address this gap by systematically exploring data-driven bottlenecks that currently hinder RLHF performance scaling, focusing specifically on the challenges posed by reward hacking and decreasing response diversity.
> To mitigate reward hacking, we introduce a hybrid reward system combining reasoning task verifiers (RTV) and a generative reward model (GenRM). This approach enables accurate assessment of responses against clearly defined ground-truth solutions. Additionally, in order to ensure response diversity and enhance learning effectiveness, we propose a novel prompt-selection method named Pre-PPO, explicitly identifying training prompts that are inherently challenging and thus less prone to reward hacking. Furthermore, we find that prioritizing mathematical and coding tasks during the early phases of RLHF training significantly boosts performance, given that these tasks naturally encode fine-grained response distinctions and possess clearly defined ground truths.
> Through experiments conducted on both small and large models, we demonstrate the effectiveness and scalability of our proposed methods. Our approach exhibits robust generalization capabilities, especially on challenging and out-of-distribution tasks, while yielding significant improvements in mathematics-intensive (STEM) and coding domains. Moreover, the proposed strategies enable the model to effectively capture subtle, task-specific distinctions during the RLHF process, substantially enhancing overall model performance. This work emphasizes the critical role of careful data construction and provides practical methodologies for addressing key performance bottlenecks in RLHF."
>
> # Broader Impact
> ## Coverage of Tasks Requiring Long-Form Outputs:
>
> While the current study primarily focuses on tasks involving short-form outputs, we have initiated experiments targeting long-form content generation. In our forthcoming paper, we employ the same methodological framework to address long-text generation tasks. Initial findings are promising, indicating that our approach scales well to long-cot setting. We believe that integrating these additional evaluations will provide a more comprehensive understanding of the applicability and robustness of our method in generating extensive textual content.
>
> ## Additional Remarks:
>
> Our approach shares similarities with concurrent work, specifically Deepseek-R1, in several aspects. Both methodologies involve initial training on verifiable tasks such as code and mathematics, followed by a preference learning phase. However, our research uniquely identifies and demonstrates the efficacy of this approach in the context of short-text generation. This distinction underscores the versatility and potential of our method across different text lengths and complexities.

---

> > ### Comment · Reviewer_UPWF · 2025-08-03
> > **Official Comment by Reviewer UPWF**
> >
> > Thank you for your rebuttal and for providing the additional experiment results, including the revised abstract, public benchmark scores, and supplementary task ordering details. We appreciate the effort to address clarity issues in the abstract and to share more insights into your mixed approach for math and coding tasks. Overall, your work demonstrates solid practical relevance in RLHF data scaling, and the new data helps clarify some aspects of your method's performance. However, the rebuttal does not fully resolve key concerns from my previous review. We respect your solid theoretical and empirical results overall, but we also have some doubts.
> >
> > **About Q4 and R1: Inconsistent Public Benchmark Results.**
> >
> > Your public benchmark results show strong coding improvements but slight declines in math tasks. This contrasts with your internal +3.9 STEM improvements, and the rebuttal lacks explanations for these patterns, potentially indicating domain-specific limitations or benchmark artifacts. Can you explain these discrepancies, particularly why math performance drops on some benchmarks?
> >
> > **About Q3 and R2: Vague Experimental Settings in Task Prioritization.**
> >
> > The supplementary experiments (coding-first: 38.1, math-first: 37.8, mixed: 38.8) provide some insight but omit crucial details on settings (e.g., data splits, training steps, hyperparameters, and full metrics like variance or entropy). Can you provide a detailed description of these setups, including dataset sizes, training duration, and how scores were computed?
> >
> > **About Q3 and R2: Incomplete Systematic Exploration of Task Prioritization.**
> >
> > You haven't tested placing math/coding tasks later in training (a common pretraining strategy) or varying data proportions (e.g., reducing non-math/coding data to 30%) as mentioned in question 3. These omissions limit the systematic exploration needed to validate your method’s design. Could you test or justify alternatives like placing math/coding later or varying proportions (e.g., 30% vs. 70% math/coding early)? If not feasible, please justify why your current mixed approach is optimal.
> >
> > **About Q3: Impact of Training Epochs on Task Prioritization Benefits.**
> >
> > You didn't specify the number of training epochs used in these experiments. Can you clarify how many epochs your experiments ran? Additionally, if training extends over multiple epochs, does the performance benefit of prioritizing math and coding tasks early persist, or does it diminish as the model converges?
> >
> > **About Q2: Omission of Differentiation from Prior Work.**
> >
> > The rebuttal ignores the request for comparisons with works like DeepSeek-R1 (2025), which shares similarities in prioritizing reasoning tasks via multi-stage RL. Without this, novelty in your prioritizing in early math/code training remains unclear. Can you provide a detailed comparison highlighting theoretical or empirical differences?
> >
> > **About R1 and R2: Insufficient Evidence for Internal Datasets.**
> >
> > Your claims about the superiority of internal evaluation sets lack quantitative support, and your supplementary experiments still rely exclusively on these internal sets. Can you substantiate the advantages of your internal datasets by comparing their characteristics with those of public benchmarks like GSM8K or HumanEval(such as providing a case study)?

---

> > > ### Author Response · Authors · 2025-08-04
> > >
> > > # Inconsistent Public Benchmark Results and Insufficient Evidence for Internal Datasets
> > >
> > > Based on your suggestion, we have expanded our evaluation to include additional challenging math benchmarks as well as Human Eval and GSM8k:
> > >
> > > | Task Series    | Baseline Score | Our Model Score |
> > > |----------------|----------------|-----------------|
> > > | Ominimath      | 26.7           | 28.0            |
> > > | OlympiadBench  | 64.7           | 68.0            |
> > > | Human Eval      | 92.1           | 92.7            |
> > > | GSM8k          | 94.2           | 94.0            |
> > >
> > > These results demonstrate our method's strong performance on medium to hard math problems. However, we acknowledge that for easier test sets (e.g., Math-500) or extremely challenging tests (e.g., AIME), our improvements are less pronounced in short-term responses.
> > >
> > > Regarding GSM8k and Human Eval, our pre-training process includes an annealing phase with substantial open-source SFT data, resulting in high baseline scores that make performance differences less discernible. We focused on other benchmarks to better highlight our method's effects.
> > >
> > >
> > > # Vague Experimental Settings in Task Prioritization
> > >
> > > For both the "code first" and "math first" approaches, we employed a three-stage training process:
> > >
> > > "Code first" approach:
> > >  Stage 1: Training on all verifiable code data;
> > >  Stage 2: Training on all math data;
> > >  Stage 3: Training on all remaining data;
> > >
> > > "Math first" approach:
> > > Stage 1: Training on all math data;
> > > Stage 2: Training on all verifiable code data;
> > > Stage 3: Training on all remaining data;
> > >
> > > In both scenarios, we adhered to the experimental setup outlined in Section 4.1 of our paper, with the following specifications:
> > >
> > > - Total training steps: 3,500
> > >   - Each step generates 4,096 samples
> > >   - Training occurs in 8 sub-steps, with each sub-step processing 512 samples
> > > - Total training data: 50 million prompts
> > > - Evaluation metric: TestSet V1.0
> > > - Metric calculation and methods: As detailed in Section 4.1
> > >
> > >
> > > # Incomplete Systematic Exploration of Task Prioritization
> > >
> > > We conducted a series of systematic explorations of task prioritization using a small model (25B) on Evaluation dataset V1.0. The experimental settings are consistent with those detailed in Section 4.1 of the paper.
> > >
> > > | First Stage | Second Stage | Final Stage | Performance |
> > > |-------------|--------------|-------------|-------------|
> > > | 50% math data and 50% verifiable code data | 50% math data and 50% verifiable code data | 100% other data | 38.2 |
> > > | 100% verifiable code data | 100% math data | --- | 38.1 |
> > > | 100% math data | 100% verifiable code data | --- | 37.8 |
> > > | 80% verifiable code data and 20% math data | 20% verifiable code data and 80% math data | --- | 38.4 |
> > > | Our SOTA method (Data proportion shown in Figure 8 of the paper) | Our SOTA method (Data proportion shown in Figure 8 of the paper) | --- | 38.8 |
> > >
> > > Furthermore, instead of using stepped curves, we employed smoother curves (e.g., sine curves), but the results were similar. Consequently, we didn't expend additional computational resources on conducting ablation studies.
> > >
> > > Our key insight is that the coding task is verifiable, while for the math task, we still used GenRM without an absolutely verifiable method, making it more susceptible to hacking. Therefore, it's a reasonable choice to train the less hackable coding task first, followed by the math task.
> > >
> > > We mixed coding and math training because we believed that combined training would boost the performance of both. This can be observed when comparing with the "code first" strategy (100% verifiable code data in the first stage and 100% math data in the second stage).
> > >
> > >
> > >
> > >
> > > # Impact of Training Epochs on Task Prioritization Benefits.
> > > We have consistently observed that as the number of training epochs increases, performance on math and coding tasks shows marked improvement. However, this positive trend is offset by a decline in overall evaluation metrics, primarily due to reward hacking in more subjective domains like creative writing. In response to this phenomenon, we have strategically shifted our focus to prioritize training on math and coding tasks. These areas have demonstrated greater resilience against hacking attempts, making them more reliable indicators of genuine model improvement.
> > >
> > >
> > > # Omission of Differentiation from Prior Work.
> > > Temporal Context and Focus:
> > > Our work and DeepSeek-R1 developed concurrently, with different focuses: we emphasized short-form RL training, while DeepSeek-R1 concentrated on long-form RL. Both approaches prioritize less hackable tasks initially (coding and math tasks), aiming to enhance RLHF effectiveness.
> > >
> > > Novel Data Selection Approach:
> > > We suggest that a core innovation in our work is the Pre-PPO approach for selecting challenging data. This method has shown promise not only in short-form RL, as demonstrated in this paper, but also in our subsequent work on long-form RL. We believe this data selection strategy represents a meaningful contribution to the field.

---

> > > > ### Comment · Reviewer_UPWF · 2025-08-08
> > > > **Official Comment by Reviewer UPWF**
> > > >
> > > > I have carefully reviewed the authors' rebuttal and appreciate the thoughtful efforts to address my concerns. However, I still believe that the differences in focus between long-form and short-form RL do not adequately address my concerns regarding the novelty of the proposed method. Furthermore, the issue of lacking open-source verification (models and evaluation benchmarks) persists, which limits the generalizability and reproducibility of the approach presented in this paper.

---

### Official Review · Reviewer_TLTJ · 2025-07-03

**Clarity:** 3
**Significance:** 3
**Originality:** 3
**Rating:** 5
**Confidence:** 4

**Summary:**

The paper presents a comprehensive study on the data scaling in the RLHF stage of LLMs. The authors study two types of rewards, i.e. GenRM and RTV (reasoning task verifier). The authors identify that simply scaling the number of prompts does not leads to better performance. Instead, the authors provide two useful strategies for data selection in RLHF:
1. Pre-PPO prompt selection. Only keep the lowest-scoring 10% prompts, which suffers less from reward hacking.
2. Prioritize math and code in early training.

The authors conduct extensive experiments on two kinds of MoE models (25B and 150B), showing that these strategies mitigate the reward hacking and the diversity deterioration problems, and effectively improve the model performance.

**Questions:**

1. Does this study cover tasks that need long-form outputs, for example, tasks (e.g., AIME) that may need inference scaling to over 10k tokens responses? If not, what would be different if these were included?

2. What if the GenRM is re-trained for several iterations during the RL process? Will that alleviate the reward hacking problem? Will the current discoveries still hold?

**Ethical Concerns:**

["NO or VERY MINOR ethics concerns only"]

**Limitations:**

yes

**Paper Formatting Concerns:**

no concern

**Quality:**

4

**Strengths And Weaknesses:**

Strength:
1. Data selection is a crucial problem that needs more research. The paper has strong motivation and addresses a core problem.
2. The experiments are very comprehensive and large-scale. The experiments need a lot of resources to be run. The information from this paper is valuable.
3. The strategies are simple but effective.

Weakness:
1. All models and benchmarks are internal, making external comparison difficult.
2. It would be good to show more details,  like models, verifiers, data.

---

> ### Author Rebuttal · Authors · 2025-07-31
>
> # Reproducibility and Use of Standard Public Benchmarks
>
> We appreciate the importance of using standard public benchmarks and have indeed conducted evaluations on several open benchmarks. Our results show:
>
> - Significant improvement in coding tasks (e.g., LiveCodeBench: 30.3 -> 36.2).
> - Slight improvements or comparable performance in general knowledge tests (e.g., MMLU: 87.3 -> 87.3, MMLU pro: 79.6 -> 80.1).
> - Slightly lower but comparable performance on simple math tasks (e.g., AMC23: 68.8 -> 68.7, MATH500: 88.2 -> 87.8).
> - Comparable performance on high-difficulty math tasks (e.g., AIME: 18.6 -> 17.9, 26.9 -> 27.2).
>
> However, we've identified limitations in many public benchmarks:
> - Potential IID issues, not fully reflecting real-world challenges.
> - Risk of data leakage due to widespread use.
> - Insufficient difficulty to comprehensively assess advanced models.
>
> For instance, our model can score nearly 90 points on some math benchmarks, but only around 30 points on our more challenging internal test sets.
>
> Consequently, we developed an internal evaluation suite with the following advantages:
> - Higher difficulty, better differentiating top-tier model capabilities.
> - Continuous iteration, preventing model overfitting.
> - Black-box testing, ensuring fair and authentic evaluation.
>
> We believe combining public benchmarks with rigorous internal assessments provides a more comprehensive and reliable evaluation of model performance. We will continue to monitor and use public benchmarks while refining our internal evaluation system.
>
> Regarding the use of open-source models and datasets, we acknowledge their value in enhancing reproducibility. We are considering conducting supplementary experiments with these resources and will include them in future work to provide a more comprehensive comparison.
>
> | Task Series       | Sub-Task Name                              | Question Count | Baseline Score | Our Model Score |
> |-------------------|-------------------------------------------|----------------|----------------|-----------------|
> | MATH - Aligned    | AMC23 (AMC23fblshot_v2.rwq_score)         | 40             | 68.8           | 68.7            |
> | MATH - Aligned    | MATH500                                   | 500            | 88.2           | 87.8            |
> | AIME (Open-Style) | AIME 2025                                 | 30             | 18.6           | 17.9            |
> | AIME (Open-Style) | AIME 2024                                 | 30             | 26.9           | 27.2            |
> | Coding - Aligned  | LiveCodeBench                             | 131            | 30.3           | 36.2            |
> | General Knowledge | MMLU                                      | 150            | 87.3           | 87.3            |
> | General Knowledge | MMLU pro                                  | 1000           | 79.6           | 80.1            |
> | Reasoning         | GPQA diamond                              | 198            | 59.6           | 60.1            |
> | Reasoning         | SuperGPQA (superGPQAfrdom_1K_sample_2)    | 250            | 52.8           | 54.0            |
>
> #  Experimental Design and Clarity
>
> We appreciate the request for more clarity on our experimental setup and exploration of alternative approaches. While computational constraints limited our ability to exhaustively test all possible configurations, we did conduct several key experiments to explore task ordering and data proportions:
>
> 1. On our 25B model, we tested alternative task orderings by:
>    a) Training exclusively on coding data first
>    b) Training exclusively on math data first
>    c) Using our current mixed approach
>
> Results on our internal test set V1 showed:
> - Coding first: 38.1
> - Math first: 37.8
> - Current mixed approach: 38.8
>
> This demonstrates that our current mixed approach outperforms exclusively prioritizing either coding or math tasks.
>
> 2. We also experimented with data composition by incorporating all verifiable reasoning data into the math dataset. This yielded nearly identical overall performance (50.6 vs 50.8), suggesting our current data balance is effective.
>
> 3. While we didn't test placing math and coding at the very end of training due to computational constraints, we explored a similar concept in subsequent long-chain-of-thought reinforcement learning experiments. These showed comparable performance on open reasoning benchmarks.
>
> These experiments provide evidence supporting our current task ordering and data balance. However, we acknowledge that further systematic exploration could yield additional insights. We're open to conducting more extensive experiments in future work, including generating per-task learning curves for our Early Training Emphasis strategy, as computational resources allow.
>
> In addition, our subsequent paper explains more details on how we implement GenRM and the specifics of PPO training. Due to anonymity policy issues, we cannot share these details at present. The final version will reference this paper.
>
> Q：What if the GenRM is re-trained for several iterations during the RL process? Will that alleviate the reward hacking problem?
>
> A：We have indeed experimented with re-training GenRM for several iterations during the RL process. This approach showed some effectiveness in alleviating the reward hacking problem, and many of our current discoveries still hold true. However, we found that this method requires significant human resources and computational power. Due to these high costs, we ultimately decided against implementing it in large-scale updates and enterprise scenarios. While it offers benefits, the trade-off in terms of resource allocation and efficiency led us to explore other, more scalable solutions for our production environments.
>
> # Broader Impact
> ## Coverage of Tasks Requiring Long-Form Outputs:
>
> While the current study primarily focuses on tasks involving short-form outputs, we have initiated experiments targeting long-form content generation. In our forthcoming paper, we employ the same methodological framework to address long-text generation tasks. Initial findings are promising, indicating that our approach scales well to long-cot setting. We believe that integrating these additional evaluations will provide a more comprehensive understanding of the applicability and robustness of our method in generating extensive textual content.
>
> ## Additional Remarks:
>
> Our approach shares similarities with concurrent work, specifically Deepseek-R1, in several aspects. Both methodologies involve initial training on verifiable tasks such as code and mathematics, followed by a preference learning phase. However, our research uniquely identifies and demonstrates the efficacy of this approach in the context of short-text generation. This distinction underscores the versatility and potential of our method across different text lengths and complexities.

---

> > ### Comment · Reviewer_TLTJ · 2025-08-04
> >
> > Thanks authors for opening more details and address my comments. I will keep my scores

---

### Official Review · Reviewer_V9Wr · 2025-07-03

**Clarity:** 2
**Significance:** 2
**Originality:** 2
**Rating:** 3
**Confidence:** 4

**Summary:**

This paper investigates the data scaling challenge in Reinforcement Learning from Human Feedback (RLHF), where increasing the amount of training data does not consistently lead to improved model performance.
To handle this issue, the authors propose the following contributions:
1) Hybrid Reward System: combining the Bradley-Terry (BT) reward model, reasoning task verifiers (RTV) and the generative reward model (GenRM).
2) Pre-PPO Prompt Selection: training with low-reward prompts to effectively mitigate the reward hacking issue
3) Early Training Emphasis: empirical finding that prioritizing mathematical and coding tasks during the early phases of RLHF training improves final performance

**Questions:**

1) Could the authors provide per-task learning curves for the Early Training Emphasis strategy? It would be helpful to see how this strategy affects individual task performance over time.

**Ethical Concerns:**

["NO or VERY MINOR ethics concerns only"]

**Final Justification:**

The authors have added further analysis on the ablation study of the reward models and the Early Training Emphasis strategy, provided additional empirical results on PrePPO, and improved the overall writing.
However, I still believe that the lack of theoretical support and novelty remains a concern.

**Limitations:**

Yes

**Quality:**

2

**Strengths And Weaknesses:**

Strengths:
1) The proposed methods are simple, easy to implement, and demonstrate clear effectiveness in the experimental results.
2) Addressing the data scaling issue in RLHF is both important and relatively underexplored, making this work a valuable contribution to the field.

Weaknesses:
1) The paper does not clearly demonstrate the individual or combined advantages of using the BT reward model, GenRM, and RTV. An ablation study or comparative analysis would strengthen this claim.
2) Key aspects of the experimental setup are under-specified. For example, the composition and evaluation methodology of the "V1.0" and "V2.0" datasets are not clearly described.
3) While the empirical results are promising, the paper lacks theoretical analysis or intuition explaining why Pre-PPO prompt selection and early prioritization of math and coding tasks improve robustness against reward hacking.
4) The writing could be improved for clarity and conciseness. For instance, the abstract contains redundant descriptions of the Pre-PPO method (e.g., line 14).
5) While the integration of components is thoughtful, the overall novelty is limited, and the paper would benefit from a more refined presentation and deeper analysis.

---

> ### Author Rebuttal · Authors · 2025-07-31
>
> # The Individual or Combined Advantages of Using the BT Reward Model, GenRM, and RTV
> We appreciate the reviewer's concern regarding the demonstration of individual and combined advantages of the BT reward model, GenRM, and RTV. We would like to clarify that we have indeed conducted a comprehensive comparative analysis of these components in the Appendix under "Further Experimental Analysis." This analysis provides substantial evidence for the effectiveness and distinctions between these methods.
>
> Specifically, our analysis focuses on two critical aspects:
>
> 1. Reward Hacking:
> We demonstrate that RTV is significantly more robust against reward hacking compared to other methods. Our experiments show a clear hierarchy in performance:
>    RTV > GenRM with ground truth > GenRM without ground truth ≈ BT RM.
> This hierarchy illustrates the superior ability of RTV to maintain integrity in reward signals, crucial for reliable model training.
>
> 2. Fine-grained Difference Detection:
> Our analysis reveals that RTV excels in identifying subtle differences between responses, outperforming other methods. The performance ranking follows the same pattern:
>    RTV > GenRM with ground truth > GenRM without ground truth ≈ BT RM.
> This capability is essential for nuanced evaluation and improvement of model outputs.
>
>
> # Test Dataset Details and Evaluation Methodology
>
> We apologize if this crucial information was not sufficiently highlighted in the main text. We will consider emphasizing these findings more prominently in the final version to ensure their significance is not overlooked. We would like to offer additional information about our test datasets:
>
> The test sets are independently created by evaluators and are divided into white-box and black-box test collections. The white-box tests are visible to algorithm developers, while the black-box tests are not. The domains covered in these datasets are already listed in the paper.
>
> Furthermore, these test sets are updated every 1-2 months. During each update, new challenging questions are added, and some simpler questions are replaced to maintain the overall difficulty level. This regular refresh ensures that the evaluation remains relevant and challenging over time.
>
> V1.0 and V2.0 refer to two consecutive versions of these test sets. This versioning reflects our commitment to continuous improvement and adaptation of our evaluation metrics.
>
> In the final version of the paper, we will include a dedicated section that elaborates on these key aspects of our experimental setup, providing readers with a more comprehensive understanding of our evaluation process and the robustness of our results.
>
> # Theoretical Justification and Novelty
> While there isn't much theoretical analysis, in the "Further Experimental Analysis" appendix, we've made some relevant analyses and interpretations.
> 1. Analyzing prompt filtering by Pre-PPO strategy
> We collected five responses per prompt, calculated the maximum edit distance among them, and categorized prompts into bins. Then we computed the average normalized reward model score for each bin. Edit distance can reflect the granularity of response differences (larger for coarser-grained, smaller for finer-grained).
> 2. Observations from the analysis
> Task-type differences: For tasks supervised by GenRM with ground truth (e.g., math and logic) and without ground truth (e.g., creative writing and cosplay), the normalized reward model scores show different trends as edit distance changes. The model captures coarse-grained differences easily for tasks without ground truth, and is more sensitive to fine - grained ones for tasks with ground truth.
> Effect of Pre-PPO exclusion: In the Pre-PPO strategy, we excluded certain prompts. An ablation study found that reintroducing previously excluded math and logic prompts improved overall performance marginally. This implies that learning coarse-grained patterns from creative writing and cosplay tasks can harm the scalability of RLHF data.
> Hypothesis on early task emphasis: Emphasizing math and coding tasks in early training may help the model capture fine-grained distinctions first, reducing the negative impact of learning coarse-grained patterns too early.
> 3. Reward model analysis
> Analyzing reward score differences across prompt bins, we found that GenRM with ground truth and RTV assign larger score differences in bins with smaller edit distances. GenRM without ground truth fails to show meaningful score differences in these bins. RTV shows greater ability to capture subtle response distinctions compared to GenRM with ground truth at low edit distances. Overall, reward models with ground truth or verification feedback are more sensitive to fine-grained response variations.
>
> # Experimental Design and Clarity
>
> We appreciate the reviewer's request for per-task learning curves for the Early Training Emphasis strategy. Due to our rebuttal policy, we are unable to include new figures in the PDF. However, readers can refer to Figure 4 in the original paper, specifically the first 1400 steps, to gain insights into the learning progression for math and coding tasks.
>
> In addition to the information provided in the paper, we have conducted supplementary experiments regarding the order of training for code and math tasks. On our 25B model, we tested the following configurations:
>
> 1. Training exclusively on coding data first
> 2. Training exclusively on math data first
> 3. Our current mixed approach
>
> Results on our internal test set V1 showed:
> - Coding first: 38.1
> - Math first: 37.8
> - Current mixed approach: 38.8
>
> These results demonstrate that our current mixed approach outperforms exclusively prioritizing either coding or math tasks.
>
> We hope this additional information provides further clarity on the effects of task ordering in our training strategy. We appreciate the reviewer's suggestion and will consider generating more detailed per-task learning curves for future iterations of our work, as computational resources allow.
>
>
> # Writing and Presentation
> We acknowledge the issues of redundancy and lack of clarity in our writing. We will promptly revise the abstract to eliminate redundancy and clearly convey our contributions. Moreover, we'll review and enhance the clarity and conciseness of other paper parts. Thank you for pointing these out.
>
> We present the revised abstract below, incorporating feedback on clarity and scope:
>
> "Reinforcement Learning from Human Feedback (RLHF) is essential for aligning large language models (LLMs) with human preferences and values. While recent research has primarily focused on algorithmic advancements—such as reducing computational overhead or strengthening reward models to mitigate reward hacking—the critical role of prompt-data construction and its scalability has received comparatively less attention. In this paper, we address this gap by systematically exploring data-driven bottlenecks that currently hinder RLHF performance scaling, focusing specifically on the challenges posed by reward hacking and decreasing response diversity. To mitigate reward hacking, we introduce a hybrid reward system combining reasoning task verifiers (RTV) and a generative reward model (GenRM). This approach enables accurate assessment of responses against clearly defined ground-truth solutions. Additionally, in order to ensure response diversity and enhance learning effectiveness, we propose a novel prompt-selection method named Pre-PPO, explicitly identifying training prompts that are inherently challenging and thus less prone to reward hacking. Furthermore, we find that prioritizing mathematical and coding tasks during the early phases of RLHF training significantly boosts performance, given that these tasks naturally encode fine-grained response distinctions and possess clearly defined ground truths. Through experiments conducted on both small and large models, we demonstrate the effectiveness and scalability of our proposed methods. Our approach exhibits robust generalization capabilities, especially on challenging and out-of-distribution tasks, while yielding significant improvements in mathematics-intensive (STEM) and coding domains. Moreover, the proposed strategies enable the model to effectively capture subtle, task-specific distinctions during the RLHF process, substantially enhancing overall model performance. This work emphasizes the critical role of careful data construction and provides practical methodologies for addressing key performance bottlenecks in RLHF."
>
> # Broader Impact
> ## Coverage of Tasks Requiring Long-Form Outputs:
>
> While the current study primarily focuses on tasks involving short-form outputs, we have initiated experiments targeting long-form content generation. In our forthcoming paper, we employ the same methodological framework to address long-text generation tasks. Initial findings are promising, indicating that our approach scales well to long-cot setting. We believe that integrating these additional evaluations will provide a more comprehensive understanding of the applicability and robustness of our method in generating extensive textual content.
>
> ## Additional Remarks:
>
> Our approach shares similarities with concurrent work, specifically Deepseek-R1, in several aspects. Both methodologies involve initial training on verifiable tasks such as code and mathematics, followed by a preference learning phase. However, our research uniquely identifies and demonstrates the efficacy of this approach in the context of short-text generation. This distinction underscores the versatility and potential of our method across different text lengths and complexities.

---

> ### Comment · Reviewer_V9Wr · 2025-08-08
>
> Thank you for the detailed response. I appreciate the clarifications.
>
> I will increase my score by one point. However, I still believe that the lack of theoretical support and novelty remains a concern.

---

### Official Review · Reviewer_PsAt · 2025-07-03

**Clarity:** 2
**Significance:** 3
**Originality:** 3
**Rating:** 4
**Confidence:** 4

**Summary:**

This paper addresses an understudied aspect of RLHF: the role of prompt-data construction and its scalability. The authors identify two bottlenecks preventing effective data scaling in RLHF—reward hacking and declining response diversity—and propose practical solutions. Their approach introduces a hybrid reward system combining Reasoning Task Verifiers (RTV) and Generative Reward Models (GenRM), along with a novel Pre-PPO prompt selection strategy that prioritizes challenging prompts. Additionally, they demonstrate that prioritizing mathematical and coding tasks during early RLHF training stages significantly improves performance. The methods are validated on models of different scales (25B and 150B parameters), showing consistent improvements across multiple evaluation metrics.

**Questions:**

1. Your method shows strongest improvements in mathematical and coding tasks. Have you tested whether the approach generalizes to more open-ended tasks like dialogue or creative writing where ground truth is less well-defined?
2. Public Benchmark Validation: Why weren't standard public benchmarks (e.g., HumanEval, MATH, GSM8K) used for evaluation? This would help the community better understand the practical impact of your improvements.

**Ethical Concerns:**

["NO or VERY MINOR ethics concerns only"]

**Limitations:**

Yes, the authors adequately discuss limitations in Section 5, acknowledging the resource-intensive nature of their approach, computational overhead, and uncertainty about long-form Chain-of-Thought RLHF. However, they could expand on:

- The potential bias introduced by focusing heavily on STEM tasks during early training
- How the method might perform with different reward model architectures
- The impact of their approach on model capabilities outside the evaluated domains

**Quality:**

2

**Strengths And Weaknesses:**

Strengths

1. Novel Problem Focus: While most RLHF research concentrates on algorithmic improvements, this paper uniquely addresses data construction and scaling—a critical but overlooked aspect that directly impacts practical deployment.
2. Strong Empirical Results: Consistent improvements across model scales (25B and 150B) and evaluation sets, with particularly impressive gains in STEM (+3.9) and coding (+3.2) tasks.
3. Good Rigor: The paper provides valuable insights into fine-grained vs. coarse-grained response differences and their impact on RLHF effectiveness, backed by comprehensive experiments in the appendices. Extensive appendices with ablation studies, case studies, and detailed analyses strengthen the paper's contributions.

Weaknesses

1. Limited Reproducibility: Due to proprietary data and code restrictions, the results cannot be independently verified, which is a challenge. The paper relies heavily on internal evaluation sets (V1.0 and V2.0) without comparison to widely-used public benchmarks, making it difficult to contextualize the improvements.
2. Theoretical Justification: While the empirical results are strong, the paper lacks theoretical analysis of why Pre-PPO selection and early task prioritization work, relying mainly on intuition and post-hoc explanations.

---

> ### Author Rebuttal · Authors · 2025-07-30
>
> # Evaluation and Impact
>
> We sincerely thank the reviewers for their insightful feedback and constructive suggestions. Below, we address the specific questions and concerns raised:
>
> Generalization to Open-Ended Tasks like Dialogue or Creative Writing ：
> We have conducted manual evaluations on creative writing tasks to assess the generalizability of our approach. The results indicate that our model maintains a consistent performance level comparable to the baselines. Similarly, in dialogue tasks, our model demonstrates performance that is on par with existing methods. These preliminary evaluations suggest that our approach is capable of handling more open-ended tasks effectively. However, we acknowledge that further extensive evaluations are necessary to thoroughly establish the generalization capabilities across diverse open-ended scenarios.
>
> Additional Remarks:
> Our approach shares similarities with concurrent work, specifically Deepseek-R1, in several aspects. Both methodologies involve initial training on verifiable tasks such as code and mathematics, followed by a preference learning phase. However, our research uniquely identifies and demonstrates the efficacy of this approach in the context of short-text generation. This distinction underscores the versatility and potential of our method across different text lengths and complexities.
>
> We appreciate the reviewers' suggestions to explore the generalization of our approach further. Building on this feedback, we are committed to expanding our evaluations to encompass a wider range of tasks, including more complex and lengthy text generation scenarios. These future endeavors will aim to validate and potentially enhance the applicability of our method in various real-world applications.
>
> # Reproducibility and Use of Standard Public Benchmarks
>
> We appreciate the importance of using standard public benchmarks and have indeed conducted evaluations on several open benchmarks. Our results show:
>
> - Significant improvement in coding tasks (e.g., LiveCodeBench: 30.3 -> 36.2).
> - Slight improvements or comparable performance in general knowledge tests (e.g., MMLU: 87.3 -> 87.3, MMLU pro: 79.6 -> 80.1).
> - Slightly lower but comparable performance on simple math tasks (e.g., AMC23: 68.8 -> 68.7, MATH500: 88.2 -> 87.8).
> - Comparable performance on high-difficulty math tasks (e.g., AIME: 18.6 -> 17.9, 26.9 -> 27.2).
>
> However, we've identified limitations in many public benchmarks:
> - Potential IID issues, not fully reflecting real-world challenges.
> - Risk of data leakage due to widespread use.
> - Insufficient difficulty to comprehensively assess advanced models.
>
> For instance, our model can score nearly 90 points on some math benchmarks, but only around 30 points on our more challenging internal test sets.
>
> Consequently, we developed an internal evaluation suite with the following advantages:
> - Higher difficulty, better differentiating top-tier model capabilities.
> - Continuous iteration, preventing model overfitting.
> - Black-box testing, ensuring fair and authentic evaluation.
>
> We believe combining public benchmarks with rigorous internal assessments provides a more comprehensive and reliable evaluation of model performance. We will continue to monitor and use public benchmarks while refining our internal evaluation system.
>
> Regarding the use of open-source models and datasets, we acknowledge their value in enhancing reproducibility. We are considering conducting supplementary experiments with these resources and will include them in future work to provide a more comprehensive comparison.
>
> | Task Series       | Sub-Task Name                              | Question Count | Baseline Score | Our Model Score |
> |-------------------|-------------------------------------------|----------------|----------------|-----------------|
> | MATH - Aligned    | AMC23 (AMC23fblshot_v2.rwq_score)         | 40             | 68.8           | 68.7            |
> | MATH - Aligned    | MATH500                                   | 500            | 88.2           | 87.8            |
> | AIME (Open-Style) | AIME 2025                                 | 30             | 18.6           | 17.9            |
> | AIME (Open-Style) | AIME 2024                                 | 30             | 26.9           | 27.2            |
> | Coding - Aligned  | LiveCodeBench                             | 131            | 30.3           | 36.2            |
> | General Knowledge | MMLU                                      | 150            | 87.3           | 87.3            |
> | General Knowledge | MMLU pro                                  | 1000           | 79.6           | 80.1            |
> | Reasoning         | GPQA diamond                              | 198            | 59.6           | 60.1            |
> | Reasoning         | SuperGPQA (superGPQAfrdom_1K_sample_2)    | 250            | 52.8           | 54.0            |
>
> # Theoretical Justification and Novelty
> While there isn't much theoretical analysis, in the "Further Experimental Analysis" appendix, we've made some relevant analyses and interpretations.
> 1. Analyzing prompt filtering by Pre-PPO strategy
> We collected five responses per prompt, calculated the maximum edit distance among them, and categorized prompts into bins. Then we computed the average normalized reward model score for each bin. Edit distance can reflect the granularity of response differences (larger for coarser-grained, smaller for finer-grained).
> 2. Observations from the analysis
> Task-type differences: For tasks supervised by GenRM with ground truth (e.g., math and logic) and without ground truth (e.g., creative writing and cosplay), the normalized reward model scores show different trends as edit distance changes. The model captures coarse-grained differences easily for tasks without ground truth, and is more sensitive to fine - grained ones for tasks with ground truth.
> Effect of Pre-PPO exclusion: In the Pre-PPO strategy, we excluded certain prompts. An ablation study found that reintroducing previously excluded math and logic prompts improved overall performance marginally. This implies that learning coarse-grained patterns from creative writing and cosplay tasks can harm the scalability of RLHF data.
> Hypothesis on early task emphasis: Emphasizing math and coding tasks in early training may help the model capture fine-grained distinctions first, reducing the negative impact of learning coarse-grained patterns too early.
> 3. Reward model analysis
> Analyzing reward score differences across prompt bins, we found that GenRM with ground truth and RTV assign larger score differences in bins with smaller edit distances. GenRM without ground truth fails to show meaningful score differences in these bins. RTV shows greater ability to capture subtle response distinctions compared to GenRM with ground truth at low edit distances. Overall, reward models with ground truth or verification feedback are more sensitive to fine-grained response variations.

---

### Note · Authors · 2025-08-11

Dear Area Chair and All Reviewers,

We appreciate the opportunity to address the concerns raised by the reviewers during the rebuttal phase. We understand that some reviewers have expressed doubts regarding the theoretical guarantees, novelty, and reproducibility of our work. We would like to emphasize several key points that may have been overlooked:

Novelty:

a. Our paper introduces not only the Prioritizing Mathematical and Coding Tasks strategy (which seems to be the main focus of the reviewers) but also the Pre-PPO strategy. We are the first to propose and validate the effectiveness of this Pre-PPO strategy in short-text scenarios.

b. Our work is pioneering in exploring RL data scaling for short-text scenarios, which, to the best of our knowledge, is the first of its kind.

c. Short-text and long-text scenarios differ significantly, particularly in terms of response diversity. In our research, we designed a strategy to prioritize mathematical and coding tasks in the initial training phase, and we also developed a Pre-PPO approach. Our key insight from this work is that we need to learn more fine-grained differences before diversity diminishes. This is crucial, especially in short-text scenarios where maintaining response diversity is challenging. By implementing these strategies, we aim to capture and preserve subtle variations in the model's responses, thereby enhancing overall performance and creativity in tasks such as short-form writing.

Theoretical Insights:

It appears that the reviewers may not have fully considered our "Further Experimental Analysis" chapter in the appendix. Due to space constraints, we included extensive experimental analyses and deeper insights in this section. While we acknowledge the lack of formal theoretical guarantees, we provide comprehensive analyses that offer valuable perspectives on RL data scaling.

Reproducibility:

We understand the concerns regarding reproducibility. Our experiments involve both large and small models, and data scaling is resource-intensive. While we haven't replicated these experiments on open-source models, we would like to highlight that the models employing our techniques have been successfully deployed in our company's cloud services and are currently serving a large user base. This real-world application indirectly validates the effectiveness of our method.


Thank you for your time and consideration.

Sincerely,

The Authors of Paper #7124

---

### Decision · Program_Chairs · 2025-09-17

**Decision:**

Accept (poster)

**Comment:**

This paper investigates data scaling challenges in RLHF, proposing several data-centric strategies to mitigate reward hacking and improve performance. The primary contributions are a hybrid reward system, a "Pre-PPO" method for selecting challenging prompts, and a curriculum learning strategy that prioritizes math and coding tasks during early training phases, all validated on large-scale internal models.

The problem has practical relevance, and the results are convincing but reviewers raised significant concerns about the lack of reproducibility due to the use of proprietary models and internal benchmarks. Other critical issues included the limited novelty, with methods closely resembling concurrent work and prior curriculum learning approaches. The author's rebuttal provided some results on public benchmarks.

For the final version, I recommend including additional results on public benchmarks. Additionally, a thorough discussion and empirical comparison to differentiate their work from prior and concurrent approaches, particularly regarding curriculum strategies in RLHF, would be good. Overall, given strong empirical and large-scale results, I am leaning towards accepting this paper.